# Batch List-Decodable Linear Regression via Higher Moments

**Ilias Diakonikolas** [*1]  **Daniel M. Kane** [*2]  **Sushrut Karmalkar** [*3]  **Sihan Liu** [*2]  **Thanasis Pittas** [*1]

## Abstract

We study the task of list-decodable linear regression using batches. A batch is called clean if the points it contains are i.i.d. samples from an unknown linear regression distribution. For a parameter $\alpha \in (0, 1/2)$, an unknown $\alpha$-fraction of the batches are clean and no assumptions are made on the remaining batches. The goal is to output a small list of vectors at least one of which is close to the true regressor vector in $\ell_2$-norm. (Das et al., 2023) gave an efficient algorithm for this task, under natural distributional assumptions, with the following guarantee. Under the assumption that the batch size $n$ satisfies $n \geq \tilde{\Omega}(\alpha^{-1})$ and the number of batches is $m = \text{poly}(d, n, 1/\alpha)$, their algorithm runs in polynomial time and outputs a list of $O(1/\alpha^2)$ vectors at least one of which is $\tilde{O}(\alpha^{-1/2}/\sqrt{n})$ close to the target regressor. Here we design a new polynomial-time algorithm for this task with significantly stronger guarantees under the assumption that the low-degree moments of the covariates distribution are Sum-of-Squares (SoS) certifiably bounded. Specifically, for any constant $\delta > 0$, as long as the batch size is $n \geq \Omega_\delta(\alpha^{-\delta})$ and the degree-$\Theta(1/\delta)$ moments of the covariates are SoS certifiably bounded, our algorithm uses $m = \text{poly}((dn)^{1/\delta}, 1/\alpha)$ batches, runs in polynomial-time, and outputs an $O(1/\alpha)$-sized list of vectors one of which is $O(\alpha^{-\delta/2}/\sqrt{n})$ close to the target. That is, our algorithm achieves substantially smaller minimum batch size and final error, while achieving the optimal list size. Our approach leverages higher-order moment information by carefully combining the SoS paradigm interleaved with an iterative method and a novel list pruning procedure for this

setting. In the process, we give an SoS proof of the Marcinkiewicz-Zygmund inequality that may be of broader applicability.

## 1. Introduction

In several modern applications of data analysis, including federated learning (Wang et al., 2021), sensor networks (Wax & Ziskind, 1989), and crowdsourcing (Steinhardt et al., 2016), it is typically infeasible to collect large datasets from a single source. Instead, samples are collected in batches from multiple sources. Unfortunately, it is often hard to find sources that provide many samples, i.e., that have large-size batches. A standard example is a movie recommendation system using rates collected from users. Here, an individual user is often unlikely to provide rating scores for a large number of movies, frequently resulting in data batches with relatively small sizes. Even less favorable, in such crowdsourcing settings, it is also the case that a *majority* of the participants might be unreliable (Steinhardt et al., 2017; 2016; Charikar et al., 2017). Such practical settings serve as motivation for this work. Formally, we study the task of linear regression under the assumption that we are given access to the model through small batches of samples collected from different sources. Importantly, as motivated by our running example, we consider the setting where *most* batches might not be collected from reliable sources.

**Definition 1.1** (List-Decodable Linear Regression using Batches)**.** Let $D_{\beta^*}$ be the distribution on pairs $(X, y) \in \mathbb{R}^{d+1}$ such that $y = {\beta^*}^\top X + \xi$, for $\xi \sim \mathcal{N}(0, \sigma^2)$ and $X \sim \mathcal{G}$ that are drawn independently from each other. Suppose we are given $m$ batches of size $n$ each, where for each batch, with probability $\alpha$ the batch consists entirely of i.i.d. samples from $D_{\beta^*}$ and with probability $1 - \alpha$ it is drawn from some arbitrary distribution. The goal is to output a list $L$ of vectors in $\mathbb{R}^d$ with $|L| \leq O(1/\alpha)$, and the guarantee that there is a $\widehat{\beta} \in L$ such that $\|\widehat{\beta} - \beta^*\|_2$ is small.

For the vanilla setting of linear regression, with batch size $n = 1$ and no outliers, the classical least-squares estimator is essentially optimal. Unfortunately, even a single outlier is enough to force the least-squares estimator to deviate arbitrarily. To address this discrepancy, Huber (1964); Rousseeuw & Leroy (1987) proposed classical robust estimators that could handle a constant fraction of outliers.

---

[*]Equal contribution [1]Department of Computer Science, University of Wisconsin Madison, Madison, United States [2]Department of Compuetr Science and Engineering, University of California San Diego, San Diego, United States [3]Microsoft Research, Cambridge, England. Correspondence to: Thanasis Pittas <pittas@wisc.edu>.

*Proceedings of the 42$^{nd}$ International Conference on Machine Learning*, Vancouver, Canada. PMLR 267, 2025. Copyright 2025 by the author(s).

However, these estimators are computationally intractable (i.e., have runtime exponential in the dimension). Starting with the works of Diakonikolas et al. (2016); Lai et al. (2016), there have been a flurry of results for efficient estimators which are robust to a small constant fraction of outliers in the data. See Diakonikolas & Kane (2023) for an overview of this field.

The regime where a *majority* of the data might be outliers, known as list-decodable setting, was initially examined for mean estimation, where Charikar et al. (2017) demonstrated the first polynomial-time algorithm. Their algorithm computes a small list of hypotheses with the guarantee that one element in the list is close to the target. Generating a list of candidates, as opposed to a single solution, is information-theoretically necessary (intuitively because the outliers can mimic legitimate data points). The size of the list typically scales polynomially with the inverse of the inlier fraction, $\alpha$. The problem of list-decodable linear regression (with batches of size $n = 1$) was first studied in Karmalkar et al. (2019); Raghavendra & Yau (2020). Unfortunately, the algorithms obtained in both of these works had sample and computational complexities scaling *exponentially* in $1/\alpha$, specifically of the form $d^{\text{poly}(1/\alpha)}$ for $d$ dimensions. Interestingly, it was subsequently shown (Diakonikolas et al., 2021) that such a dependence may be inherent (for Statistical Query algorithms and low-degree polynomial tests — two powerful, yet restricted, models of computation).

Motivated by this hardness result, Das et al. (2023) proposed the batch version of the problem (Definition 1.1). The hope was that by introducing (sufficiently large) batches, the exponential complexity dependence on $1/\alpha$ can be eliminated. Before we summarize their results, some comments are in order regarding Definition 1.1. First, in the extreme case where the batch size is $n = 1$, we recover the standard list-decodable setting. Second, in the other extreme where $n = \Omega(d)$, the problem becomes straightforward, since one batch contains sufficient information to recover the target regression vector. As discussed in our running example, the batch size, which corresponds to the number of samples collected from a single source, is rarely large enough in real-world applications with high-dimensional data. This leaves the regime of $1 < n \ll d$ as the most meaningful. Das et al. (2023) showed that using $m = \text{poly}(d, n, 1/\alpha)$ batches of size $n \geq \tilde{\Omega}(1/\alpha)$, it is possible to efficiently recover a list of size $O(1/\alpha^2)$ containing an element $\hat{\beta}$ with $\|\hat{\beta} - \beta\|_2 = O(\sigma/\sqrt{n\alpha})$. Their algorithm runs in fully polynomial time, thus escaping the exponential dependence on $1/\alpha$.

The linear dependence on $1/\alpha$ in the minimum batch size $n$ is inherent in the approach of Das et al. (2023). Motivated by the practical applications of the batch setting, here we ask whether efficient algorithms are possible that succeed with significantly smaller batch size and/or with better error guarantees:

*Is there a computationally efficient algorithm for list-decodable linear regression in the batch setting with significantly improved batch size and/or error guarantees?*

Here we answer this question in the affirmative. In particular, we provide an algorithm that for any constant $\delta > 0$, it runs in polynomial time and succeeds with minimum batch size $n = \Theta_\delta(\alpha^{-\delta})$ achieving error $O_\delta(\sigma\alpha^{-\delta/2}/\sqrt{n})$. As a note regarding notation, we will switch from using the parameter $\delta > 0$ to using $k = \lceil 1/\delta \rceil$ throughout the paper.

## 1.1. Our Results

Throughout our work, we assume that the clean covariate distribution satisfies the following conditions.

**Assumption 1.2.** Let $X$ be the clean covariates distribution from Definition 1.1. We assume that

1. $X$ is $L_4$-$L_2$ hypercontractive, i.e., for any $u \in \mathbb{R}^d$, it holds $\mathbf{E}\left[(u^\top X)^4\right] \leq O(1) \ \mathbf{E}\left[(u^\top X)^2\right]^2$.
2. $X$ has identity second moment, i.e., $\mathbf{E}[XX^\top] = \mathbf{I}$.
3. There exists $Q \geq 1$ [1] and an integer $\Delta$ such that for all integer $t \in [\Delta]$ the degree-$2t$ moments of $X$ are SoS certifiably bounded by $Q$ (see Definition 3.2 for the formal definition).

We note that assumptions 1 and 2 are common in the context of robust linear regression (see, e.g., Das et al. (2023); Cherapanamjeri et al. (2020)). Assumption 3 is made so that the algorithm can take advantage of higher-order moment information from the distribution and is satisfied by a wide range of structured distributions, e.g., all strongly logconcave distributions.

Our main result is the following theorem.

**Theorem 1.3** (Main Algorithmic Result). *Let $\alpha \in (0, 1/2)$, $\sigma > 0$, $k \in \mathbb{Z}^+$ and $\beta^* \in \mathbb{R}^d$. Assume that $\sigma \leq R$, $\|\beta^*\|_2 \leq R$, and $k \leq \Delta/2$. There is an algorithm that takes as input $\alpha, \sigma, R, k$, draws $m = \tilde{O}\left(\left((kd)^{O(k)}/\alpha + \alpha^{-3}\right) \log\left(\frac{R}{\sigma}\right)\right)$ batches of size $n = O(k^2 Q^{2/k} \alpha^{-6/k})$ from the distribution of Definition 1.1, and returns a list of estimates of size $O(\alpha^{-1})$, such that, with high probability, there exists at least one estimate $\hat{\beta}$ satisfying $\|\hat{\beta} - \beta^*\|_2 = O\left(kQ^{1/k}\sigma\alpha^{-3/k}/\sqrt{n}\right)$.*

Some remarks are in order. Theorem 1.3 provides a substantial qualitative improvement over the bounds of Das et al. (2023) by succeeding for a dramatically smaller batch size while at the same time improving the estimation error. Concretely, our algorithm can use batch size of $n = O(k^2\alpha^{-6/k})$ for any $k \in \mathbb{Z}_+$ of our choice, while Das et al. (2023) was only able to work with $n = \Omega(1/\alpha)$. We note that these improvements are possible due to our

---

[1] Since $X$ has identity covariance, this implies that the bound $Q$ has to be at least 1.

stronger distributional assumptions that allow us to leverage higher moments.

Conceptually, we view the capability of our algorithm to work with a *flexible* batch size as a valuable feature—especially in real-world applications where the batch size corresponds to quantities that are not controllable by algorithm designers, i.e., the number of datapoints contributed by each provider. Our result essentially shows that there is a smooth tradeoff between the batch size provided and the computational resources required. More generally, our algorithm can cover the entire regime of $C \log^2(1/\alpha) \leq n \leq C/\alpha$, if we do not necessarily restrict $k$ to be an absolute constant. A limitation is that reaching the lower end of the regime would require $k$ to be super-constant, namely $k \sim \log^2(1/\alpha)$, which would result in quasi-polynomial runtime. Interestingly, even for that lower regime of $n$, Theorem 1.3 gives the first non-trivial (i.e., sub-exponential time) algorithm for the task.

Complementing our upper bounds, we point out that the super-polynomial dependence for extremely small values of $n$ might be inherent. Via a simple reduction from the non-batch to the batch-setting (combined with the lower bound of Diakonikolas et al. (2021)), we give evidence that the computational resources used in the algorithm of Theorem 1.3 do not suffice for $n$ significantly smaller than $\log(1/\alpha)$. See Appendix F for the relevant discussion.

### 1.2. Technical Overview

**Prior Techniques** We start by reviewing the algorithm of Das et al. (2023), which uses a batch size of $n = \tilde{\Omega}(1/\alpha)$. For simplicity, consider the case where the covariates are standard normal. The high-level idea in Das et al. (2023) is to search for approximate stationary points [2] of the $L_2$-loss, $f(\beta) = \frac{1}{2} \ \mathbf{E}_{(X,y)} \left[ (\beta^\top X - y)^2 \right]$. Without outliers, the expected gradient precisely equals $\beta - \beta^*$. If we recover a $\xi$-approximate stationary point for the inlier distribution, we can estimate $\beta^*$ up to an error of $O(\xi)$, given enough samples. With outliers, the method exploits an upper bound on the covariance of the gradient distribution of the inliers to detect outliers and to control their influence. They then use the multi-filter approach for list-decodable estimation (Diakonikolas et al., 2020) to find a subset of the samples with the covariance matrix of the gradients being upper bounded by $O(1) \|\beta - \beta^*\|_2^2/n\mathbf{I}$, with an $\alpha$ overlap with the inliers. This ensures that a $\xi$-approximate stationary point under the corrupted sample distribution will still be a $(\xi + \|\beta - \beta^*\|_2/\sqrt{n\alpha})$-approximate stationary point under the inlier distribution. This means that $\beta$ approximates $\beta^*$ up to an error of $\xi + \|\beta - \beta^*\|_2/\sqrt{n\alpha}$. Unfortunately, this results in $\|\beta - \beta^*\|_2 \leq O(\xi)$ *only when* $n \gg 1/\alpha$, since

---

[2] A vector $\beta \in \mathbb{R}^d$ is called a $\xi$-approximate stationary point of some function $f : \mathbb{R}^d \mapsto \mathbb{R}$ if it holds $\|\nabla f(\beta)\|_2 \leq \xi$.

$\sqrt{\alpha n}$ needs to be larger than 1.

In the remainder of this section, we outline the ideas of our approach.

**Iterative Estimation of the Regressor** Our main idea is to incorporate higher moment information into the estimator to alleviate the requirement on batch sizes. There are two main challenges in exploiting higher moment information. First, existing multi-filter approaches in the literature do not exploit higher moments, thus they are not easily modified. Second, existing higher-moment filters are not designed for iterative use. They can generate a list of potential candidates but cannot progressively refine sample clusters for cleaner data segmentation, as the gradient descent method by Das et al. (2023) requires.

To address this, our approach adopts a similar framework to Diakonikolas et al. (2019) for robust linear regression with a small fraction of outliers, but in a non-batch setting. Here is an overview of their algorithm: They begin by estimating the mean of the product of the covariate $X$ with the label $y$. In the outlier-free setting, this expectation equals to the true regressor $\beta^*$, and the covariance matrix can be bounded above by $O\left(\|\beta^*\|_2^2\right)\mathbf{I}$ (assuming $\|\beta^*\|_2 \gg \sigma$). They then use an algorithm for robust mean estimation for bounded-covariance distributions to derive an initial estimate $\hat{\beta}$ with a bounded error relative to $\beta^*$. They then improve the error by bootstrapping this approach. To do this, they adjust the labels via the transformation $y' = y - \hat{\beta}^\top X$. This reduces the problem of learning $\beta^*$ to another robust linear regression instance whose solution has much smaller norm. Repeating this process iteratively allows them to refine their estimate to a final error of $O(\sqrt{\epsilon}\sigma)$.

In our setting, a natural strategy is to replace the robust mean estimation algorithm with one designed for list-decodable mean estimation, since the fraction of corruptions is larger than $1/2$. Suppose that we have an algorithm $\mathcal{A}$ which produces a list of candidate regressors $\{\beta_i\}_{i=1}^m$ such that at least one of them satisfies $\|\beta_i - \beta^*\|_2 \leq \|\beta^*\|_2/2$. By applying the transformation $y' = y - \beta_i^\top X$, we can create $m$ distinct linear regression instances such that the regressor of one of these will have a significantly smaller norm. This allows us to compute more accurate estimates in the next iteration.

**Beyond Second Moments** As mentioned in the last paragraph above, the natural approach is to iteratively use a list-decodable mean estimation algorithm like the one from Diakonikolas et al. (2020) in order to estimate the mean of the random variable $W = \frac{1}{|B|} \sum_{(X,y) \in B} yX$ (where $B$ is an inlier batch) and reduce the error by a factor of 2 in each iteration. List-decodable mean estimators that rely only on second moment information have error behaving like $\sqrt{\|\mathbf{Cov}(W)\|_{op}}/\sqrt{\alpha}$, where $\sqrt{\|\mathbf{Cov}(W)\|_{op}} =$

$O\left(\|\beta^*\|_2/\sqrt{n}\right)$ is the maximum standard deviation of $W$ along any direction. This already reveals the problem with this approach: for the error to become less than $\|\beta^*\|_2/2$, we need batch size $n \gg 1/\alpha$.

We overcome this (Proposition 3.1) by using a list-decodable mean estimator that uses higher moment information, like Theorem 5.5 from Kothari & Steinhardt (2017), or Theorem 6.17 from Diakonikolas & Kane (2023). These algorithms are based on the Sum-of-Squares hierarchy and their guarantee is that whenever the higher moments of the inliers are "SoS-certifiably bounded" by $M$, then the estimation error is $O(M^{1/(2k)}\alpha^{-3/k})$.

However, to leverage the above SoS-based algorithm, we require sharp SoS bounds on moments of the batched regressor estimator $W = \frac{1}{|B|}\sum_{(X,y)\in B} yX$, which is a sum of i.i.d. random variables, while our distributional assumption only posits that the covariate $X$ has certifiably bounded moments. The fact that $W$ should also have bounded moments (but not necessarily SoS certifiable) follows from the famous Marcinkiewicz Zygmund Inequality. Unfortunately, to the best of our knowledge, an SoS proof of this inequality does not exist in the literature. We give the first SoS proof of the inequality using combinatorial arguments (cf. Lemma 3.3). We believe that this technical lemma may be of broader applicability.

Given the SoS moment bounds on $W$, the SoS-based list-decoding algorithm allows us to construct a list of size $O(1/\alpha)$ such that one of the estimates is $O_k(((\|\beta^*\|_2^{2k}/n^{2k})^{1/(2k)}\alpha^{-3/k})$-close to $\beta^*$. Hence, we will have some estimate $\beta_i$ satisfying $\|\beta_i - \beta^*\|_2 \le \|\beta^*\|/2$ whenever $n \gg \alpha^{-3/k}$, which is a significant relaxation from the condition $n \gg 1/\alpha$ required by both the first approach and the approach of Das et al. (2023).

**List Size Pruning** Having gotten the right estimate for one step, we bootstrap this to design an iterative algorithm such that the final list will contain an element that is sufficiently close. A significant challenge arises during the iterative phase of our list decoding algorithm. Initially, we generate a list of $O(1/\alpha)$ hypotheses, with the guarantee that at least one of them is near $\beta^*$. For each hypothesis, iterating further produces another $O(1/\alpha)$ hypotheses for each of the original hypotheses. Without careful management, this process can lead to an increase in the number of hypotheses that scales exponentially with the number of iterations, rendering the algorithm's complexity infeasible.

We overcome this (Proposition 3.6) with techniques inspired by Theorem A.1 in Diakonikolas et al. (2020) (see also Exercise 5.1 in Diakonikolas & Kane (2023)), which performs list-size reduction for list-decodable mean estimation. The general principle behind these methods is to check whether each hypothesis in the list has a $\Theta(\alpha)$-fraction subset of the samples associated with it such that the hypothesis

"explains" these samples. This results in certain "consistency" tests, on the basis of which we can prune elements of the list. The tests are designed such that (i) $\beta^*$ and the subset of inlier samples should pass the consistency tests, and that (ii) for any pair of sufficiently separated hypotheses, if they both pass the tests, their corresponding sets cannot have a large overlap. Given property (ii), the argument from Diakonikolas et al. (2020) shows that we can find a small cover of the set of plausible hypotheses.

In the list pruning step for list-decodable mean estimation, one usually leverages the fact that the inlier samples cluster closely around the learned mean (see, e.g., Diakonikolas et al. (2018; 2020)). This ensures survival of the optimal mean from the pruning procedure. One may want to generalize the test to the linear regression setting by asserting that $Xy$ should concentrate around the candidate regressor $\beta$. However, such a test turns out to be sub-optimal for linear regression [3]. Instead, we design the following "cross-candidate" test: we keep $\beta$ only if $\beta$ demonstrates a smaller empirical $\ell_2$ error for an $\alpha$-fraction of selected batches in comparison to *any other regressor significantly distant from $\beta$*. One may wonder whether the best regressor in the list can still survive the test, as there may be multiple equally good candidate regressors in the list with respect to the same cluster of batches. However, we note that the regressor is only compared to *distant* regressors. Consequently, they must all be far from the optimal regressor (by the triangle inequality), ensuring the survival of the best regressor. This is formally shown in Lemma 3.8.

### 1.3. Related Work

In this section we discuss related works from list-decodable linear regression and robust learning from batches. The problem of mixed linear regression is related very closely to our work as well, but due to space restrictions, we defer the relevant discussion to Appendix A.

**List-decodable Linear Regression** The list-decoding framework was first introduced in the context of machine learning in Charikar et al. (2017). They derived the first polynomial time algorithm for list-decodable mean estimation when the covariance is bounded. Later work considered the problem of list-decodable *linear regression* in the non-batch setting (Karmalkar et al., 2019; Raghavendra & Yau, 2020). Unfortunately the runtime and sample complexity had an exponential dependence on $1/\alpha$, this was later shown to be necessary for SQ algorithms (Diakonikolas et al., 2021).

**Robust Learning from Batches** The problem of learning discrete distributions from untrusted batches was introduced

---

[3]Intuitively, this is because such a test fails to take into account the influence of the size of $\beta$ on the concentration of the inlier samples.

in Qiao & Valiant (2018), which gave exponential-time solutions. Progress was made by Chen et al. (2020b) and Jain & Orlitsky (2020), achieving quasi-polynomial and polynomial runtimes, respectively, with the latter also obtaining optimal sample complexity. Further developments by Jain & Orlitsky (2021) and Chen et al. (2020a) expanded this work to one-dimensional structured distributions. Das et al. (2023) was the first to study the problem of list-decodable linear regression in the batch setting. Compared to Das et al. (2023), our method demonstrates substantial improvements in the error and the required batch size, when the covariates are i.i.d. samples from $\mathcal{N}(0, I)$. This can be attributed to our algorithm's ability to efficiently utilize higher moment information, allowing for smaller batch sizes of $\Omega(k\alpha^{-6/k})$ and achieving an error of $O_{k,\sigma}(\sigma\alpha^{-3/k}/\sqrt{n})$, marking a significant improvement over a batch size of $\Omega(\alpha^{-1})$ and error of $O(\sigma/\sqrt{\alpha n})$, achieved in Das et al. (2023).

**Organization** In Section 2, we define our notation and state some basic definitions about SoS programming. In Section 3, we describe the main parts of our algorithm in Sections 3.1, and 3.2; we then put things together to prove our main theorem in Section 3.3.

## 2. Preliminaries

**Notation** We use $X \sim D$ to denote that a random variable $X$ is distributed according to the distribution $D$. We use $\mathcal{N}(\mu, \Sigma)$ for the Gaussian distribution with mean $\mu$ and covariance matrix $\Sigma$. For a set $S$, we use $X \sim S$ to denote that $X$ is distributed uniformly at random from $S$. We write $a \ll b$ to denote that $\alpha \leq c \cdot b$ for a sufficiently small absolute constant $c > 0$. We use $a(n) = O_k(b(n))$ to denote that there is a constant $C$ such that for all $n > C$, $a(n) \leq C_k \cdot b(n)$ for a constant $C_k$ that can arbitrarily depend on $k$.

**Sum-of-Squares Preliminaries** The following notation and preliminaries are specific to the SoS part of this paper. We refer to Barak & Steurer (2016) for a more complete treatment of the SoS framework.

**Definition 2.1** (Symbolic Polynomial). A degree-$k$ symbolic polynomial $p$ with input dimension $d$ is a collection of indeterminates $\widehat{p}(\alpha)$, one for each multiset $\alpha \subseteq [d]$ of size at most $k$. We think of it as representing a degree-$k$ polynomial $p : \mathbb{R}^d \to \mathbb{R}$ whose coefficients are themselves indeterminates via $p(x) = \sum_{\alpha \subseteq [d], |\alpha| \leq k} \widehat{p}(\alpha)x^\alpha$.

**Definition 2.2** (SoS Proof). Let $x_1, \ldots, x_n$ be indeterminates and $\mathcal{A}$ be a set of polynomial equalities $\{p_1(x) = 0, \cdots, p_w(x) = 0\}$. An SoS proof of the inequality $r(x) \geq 0$ consists of two sets of polynomials $\{r_i(x)\}_{i \in [m]} \cup \{\bar{r}_i(x)\}_{i \in [w]}$ such that $r(x) = \sum_{i=1}^{m} r_i^2(x) + \sum_{i=1}^{w} p_i(x)\bar{r}_i(x)$. If the polynomials $\{r_i^2(x)\}_{i=1}^{m} \cup \{\bar{r}_i(x)p_i(x)\}_{i=1}^{w}$ all have degree at most $K$, we say that

this proof is of degree $K$ and write $\mathcal{A} \vdash_K r(x) \geq 0$. When we want to emphasize that $x$ is the indeterminate in a particular SoS proof, we write $\mathcal{A} \vdash_K^x r(x) \geq 0$. When $\mathcal{A}$ is empty, we omit it from the notation.

## 3. SoS Based Algorithm for List-Decodable Linear Regression with Batches

Our algorithm iteratively updates a list of candidates, ensuring that, in every iteration, at least one candidate from the list is close to the target regressor. It does so by iteratively applying two subroutines. In Subsection 3.1, we discuss a list-decoding subroutine that, given batch sample queries, generates a list of candidates containing some near-optimal regressor. In Subsection 3.2, we discuss a pruning subroutine that ensures that the size of our list remains bounded. Finally, in Subsection 3.3 we combine these components into the main algorithm (Algorithm 2) and prove our main theorem.

### 3.1. Single Iteration: Approximate Estimation of $\beta^*$

In this section, we construct an efficient SoS-based list-decoding algorithm to estimate the regressor $\beta^*$, assuming that $\|\beta^*\|_2 \leq R$. Specifically, this can be used to perform crude list-decodable estimation of the optimal regressor. In the final algorithm, we will bootstrap this method to generate our final list with improved error guarantee.

**Proposition 3.1.** *Let $\alpha \in (0, 1/2)$, $\delta \in (0, 1)$, $m, n, k \in \mathbb{Z}_+$, $\sigma, R > 0$, $\beta^* \in \mathbb{R}^d$. Assume $\|\beta^*\|_2 \leq R$ and $k \leq \Delta/2$. Then, there exists an algorithm that takes $\alpha, k, \delta, \sigma, R$ in the inputs, it draws $m = O\big((4kd)^{8k}Q^{-1} + 1\big)\alpha^{-1}\log(1/\delta)$ many batches from the corrupted batch distribution of Definition 1.1, runs in time $\mathrm{poly}(d^k m)$, and outputs $O(\log(1/\delta)\alpha^{-1})$ many estimations such that there exists at least one estimation $\hat{\beta}$ satisfying $\|\hat{\beta} - \beta^*\|_2 \leq O\big((k\sqrt{n})\, Q^{1/(2k)}\, (R + \sigma)\, \alpha^{-3/k}\big)$ with probability at least $1 - \delta$ over the randomness of the batches.*

A standard way of estimating the regressor is to consider the random variable $yX$. When there are no outliers, $yX$ gives an unbiased estimator of $\beta^*$. In the batch setting, a natural estimator is to use the batch average $Z_B := \frac{1}{n}\sum_{(X,y) \sim B} yX$. The main idea behind Proposition 3.1 is that we can leverage the property that the batch average estimator $Z_B$ has *SoS-certifiably bounded central moments* when $B$ consists of i.i.d. samples from the uncorrupted linear regression distribution $D_{\beta^*}$ (cf. Definition 1.1).

**Definition 3.2** (SoS-Certifiably Bounded Central Moments). Let $M > 0$, $k$ be an even integer, and $D$ be a distribution with mean $\mu$. We say that $D$ has $(M, k, K)$-certifiably bounded moments if $\{\|v\|_2^2 = 1\} \vdash_K^v \mathbf{E}_{X \sim D}\big[(v^\top(X - $

$\mu))^k] \leq M$. We say a set of points $T$ has $2k$-th central moments SoS-certifiably bounded by $M$ if the empirical distribution over these points does so.

Observe that $Z_B$ is the sum of $n$ i.i.d. copies of $yX$, and $X$ has SoS-certifiably bounded moments by Assumption 1.2. Applying the Marcinkiewicz-Zygmund Inequality, which controls the moments of i.i.d. random variables by their individual moments, will almost immediately yield that $Z_B$ also has bounded central moments. To further show that the bound is SoS-certifiable, we thus require an SoS version of this moment inequality, which is provided below.

**Lemma 3.3** (SoS Marcinkiewicz-Zygmund Inequality)**.** *Let* $v \in \mathbb{R}^d$, $X_1, \cdots, X_n$ *be i.i.d. random real vectors in* $\mathbb{R}^d$, *and* $p : \mathbb{R}^d \times \mathbb{R}^d \mapsto \mathbb{R}$ *be a degree-$t$ polynomial. Assume that*

$$\{\|v\|_2^2 = 1\} \left|\frac{v}{2kt}\right. \mathbf{E}\left[(p(v, X_i) - \mathbf{E}[p(v, X_i)])^k\right] \leq M$$

*for some number* $M > 0$. *Then the degree-$k$ central moment of the sum of* $p(v, X_i)$ *is also SoS-certifiably bounded:*

$$\{\|v\|_2^2 = 1\} \left|\frac{v}{kt}\right. \mathbf{E}\left[\left(\sum_{i=1}^n (p(v, X_i) - \mathbf{E}[p(v, X_i)])\right)^k\right]$$
$$\leq (kn)^{k/2} M. \tag{1}$$

Combining the above SoS inequality with the fact that $yX = \left(\beta *^\top X + \xi\right) X$ is a degree-2 polynomial in $X$, which has SoS certifiably bounded moments, and $\xi$, which follows the Gaussian distribution, then gives essentially a population version of the moment bound. The SoS moment bound on the empirical distribution over samples then follows by a careful analysis on the concentration properties of the empirical moments of $Z_B$. See Appendix D for the detailed argument.

**Lemma 3.4** (SoS Moment Bound)**.** *Let* $\alpha \in (0, 1/2)$, $\sigma > 0$, $k \in \mathbb{Z}^+$, $\beta^* \in \mathbb{R}^d$. *Let* $T$ *be a set of* $m$ *batches drawn according to the distribution* $D_{\beta^*}$ *defined in Definition 1.1, and batch size* $n$. *Assume that the clean covariates distribution* $X$ *satisfies Assumption 1.2 and* $k \leq \Delta/2$. *Define* $Z_B = \frac{1}{n} \sum_{(X,y) \in B} Xy$. *Suppose* $m \gg \left((4kd)^{8k} Q^{-1} + 1\right) \alpha^{-1}$. *Then the following holds with probability at least* 0.9: *(a)* $\{Z_B \mid B \in T\}$ *has* $(M, 2k, 4k)$-*certifiably bounded moments for some* $M = O((2k)^{2k}/n^k) Q (\sigma^{2k} + 2 \|\beta^*\|_2^{2k})$, *and (b)* $\mathbf{Cov}_{B \sim T}[Z_B] \preceq O((\|\beta^*\|_2^2 + \sigma^2)/n)\mathbf{I}$.

Once we have that the central moments of $Z_B$ are certifiably bounded, Proposition 3.1 follows from the following SoS-based list-decodable mean-estimation algorithm:

**Lemma 3.5** (Theorem 5.5 from Kothari & Steinhardt (2017))**.** *Let* $S$ *be a set of points in* $\mathbb{R}^d$ *containing a subset* $S_{good}$ *with* $|S_{good}| \geq \alpha|S|$. *Moreover, assume that* $S_{good}$ *has* $(M, 2k, K)$-*certifiably bounded moments*

*for some positive integers* $k, K$ *and* $M > 0$. *Then there exists an algorithm that, given* $S, k, K, M$ *and* $\alpha$, *runs in time* $\text{poly}(d^K, |S|)$, *and with probability* 0.9, *returns a list of* $O(1/\alpha)$ *many vectors containing some* $\hat{\mu}$ *with* $\|\hat{\mu} - \mu_{S_{good}}\|_2 = O\left(M^{1/(2k)} \alpha^{-3/k}\right)$.

*Proof of Proposition 3.1.* Suppose we take $m \gg \left((4kd)^{8k} Q^{-1} + 1\right) \alpha^{-1}$ many batches. Then $m \ \Omega(\alpha) \gg (4kd)^{8k} Q^{-1} + 1$ many of these batches are of inlier type with high constant probability. We denote the set of these batches by $G$. Define $Z_B = \frac{1}{n} \sum_{(X,y) \in B} Xy$. Let $D_{\beta^*}^{\otimes n}$ be the distribution of a clean batch of size $n$ whose samples are all i.i.d. from $D_{\beta^*}$ (cf. Definition 1.1). As shown in the proof of Lemma 3.4, we have $\mathbf{E}_{B \sim D_{\beta^*}^{\otimes n}}[Z_B] = \beta^*$ and $\mathbf{Cov}_{B \sim D_{\beta^*}^{\otimes n}}[Z_B] \preceq O\left((\sigma^2 + R^2)/n\right)\mathbf{I}$. Since $|G| \gg (4kd)^{8k} Q^{-1} + 1$, by Markov's inequality, it holds that

$$\left\|\sum_{B \in G} Z_B/|G| - \beta^*\right\|_2 \leq O((\sigma + R)/\sqrt{n}) \tag{2}$$

with high constant probability. Besides, since $|G| \gg (4kd)^{8k} Q^{-1} + 1$, Lemma 3.4 shows that $\mathcal{Z} := \{Z_B \mid B \in G\}$ has $2k$-th central moments SoS-certifiably bounded by

$$M = O((2k)^{2k}/n^k) Q (\sigma^{2k} + 2 \|\beta^*\|_2^{2k}) \tag{3}$$

with high constant probability. Moreover, the covariance of $\mathcal{Z}$ can be bounded from above by

$$\mathbf{Cov}_{Z \sim \mathcal{Z}}[Z] \preceq O\left((\sigma^2 + R^2)/n\right)\mathbf{I} \tag{4}$$

with high constant probability. By the union bound, Equation (2), Equation (3), and Equation (4) hold simultaneously with high constant probability. Conditioned on that, Lemma 3.5 thus allows us to estimate the mean of $\{Z_B \mid B \in G\}$ up to accuracy $O\left((k/\sqrt{n}) Q^{1/(2k)} (\sigma + R) \alpha^{-3k}\right)$. Our estimate is then $O\left((k/\sqrt{n}) Q^{1/(2k)} (\sigma + R) \alpha^{-3k}\right)$ close to $\beta^*$ by Equation (2), and the triangle inequality. This concludes the proof of Proposition 3.1. Finally we can boost the probability of success to $1 - \delta$ by running the above procedure $\log(1/\delta)$ many times and combining the lists obtained. $\square$

### 3.2. Pruning Routine

In this subsection, we show that there is an algorithm, Pruning, which reads a list $L$ containing a candidate close to $\beta^*$, and returns a sub-list $L' \subseteq L$ of size $O(1/\alpha)$ also containing a candidate close to $\beta^*$.

**Proposition 3.6** (Pruning Lemma)**.** *Let* $\alpha \in (0, 1/2)$, $\delta \in (0, 1)$, $k, n \in \mathbb{Z}_+$, $\sigma, R > 0$, *and* $\beta^* \in \mathbb{R}^d$. *Let* $L \subset \mathbb{R}^d$ *be a list of candidate regressors, and* $\beta \in L$ *be a regressor such that* $\|\beta - \beta^*\|_2 < R$. *Assume*

*that the batch size $n$ satisfies that $n \gg k \ Q^{2/k} \ \alpha^{-2/k}$ and $k \le \Delta/2$. Then there exists an algorithm* Pruning *that takes the list $L$, and the numbers $\alpha, \delta, R$ as input, draws $m = O\left(\min\left(\log(|L|), d^2\right) \log(1/\delta) \alpha^{-3}\right)$ many batches from the corrupted batch distribution of Definition 1.1, runs in time $\text{poly}(dm|L|)$, and outputs at most $O(1/\alpha)$ candidate regressors $L' \subseteq L$ such that there is at least one regressor $\beta \in L'$ satisfying $\|\beta - \beta^*\|_2^2 \le O\left(R + k\alpha^{-1/k}\sigma Q^{1/k}/\sqrt{n}\right)$ with probability at least $1 - \delta$ over the randomness of the batches drawn.*

The Pruning algorithm involves two phases. Initially, it filters regressors $\beta \in L$ by keeping those matching a certain set of solvable linear inequalities. Then it selects a subset of the remaining regressors, ensuring each pair is adequately distant. Lemmas 3.7 and 3.8 respectively show that the refined list is not excessively large and that it contains a vector near the true regressor $\beta^*$, if such a candidate exists in the original list $L$. The proof of Proposition 3.6 follows from the above two lemmas.

For each regressor, we now describe the set of linear inequalities used in the pruning process involving a *weighting function* $\mathcal{W}$ over the set of batches $T$. At a high level, a weighting function can be interpreted as a "soft cluster" for each candidate regressor $\beta$, and the inequalities aim to identify a soft cluster for each candidate regressor $\beta$ by ensuring: (i) at least an $\alpha$-fraction of batches are included in the cluster, and (ii) $\beta$ has a smaller empirical $\ell_2$ error in comparison to any other regressor $\beta'$ that is significantly distant from $\beta$, based on the following conditions involving the constant $c$, radius $R$, standard deviation $\sigma$, and batch size $n$. We denote this set of linear inequalities by $\text{IE}(\beta; L, T, R)$, i.e., $\text{IE}(\beta; L, T, R)$ is the following set of inequalities in the variable(s) $\mathcal{W} : T \mapsto [0, 1]$:

$$\sum_{B \in T} \mathcal{W}(B) \ge 0.9\alpha|T|, \tag{5}$$

$$\forall \beta' \in L \text{ satisfying } \|\beta' - \beta\|_2 \ge c\left(R + \frac{k\alpha^{-1/k}\sigma Q^{1/k}}{\sqrt{n}}\right)$$

for some sufficiently large constant $c$,

$$\sum_{B \in T} \mathbb{1}\{\sum_{(X,y) \in B} \left(y - X^\top\beta\right)^2$$
$$\le \sum_{(X,y) \in B} \left(y - X^\top\beta'\right)^2\} \mathcal{W}(B) \le \frac{\alpha}{20} \sum_{B \in T} \mathcal{W}(B). \tag{6}$$

We now provide some intuition about why there cannot be too many regressors whose associated linear inequalities are satisfiable subject to the constraint that they are all sufficiently separated. At a high level, this is because condition (ii) enforces the soft clusters associated with two sufficiently separated candidate regressors must have small intersection as two candidate regressors cannot *simultaneously* do better than the other in terms of their empirical errors on *the same batch*. We now precisely state the lemmas. For proofs, please see Appendix E.

**Lemma 3.7** (List Size Bound). *Let $R > 0$, and $L$ be a list of candidate regressors. Let $T$ be a set of batches. Let $L' \subseteq L$ be a sublist of candidate regressors satisfying the following conditions: (1) $IE(\beta; L, T, R)$ has solutions for each $\beta \in L'$, and (2) $\|\beta_1 - \beta_2\|_2 \ge c\left(R + k\alpha^{-1/k}\sigma Q^{1/k}/\sqrt{n}\right)$ for any two $\beta_1, \beta_2 \in L'$. Then it holds the size of $L'$ is at most $O(1/\alpha)$.*

The next lemma we need shows that the list after an application of Lemma 3.7 contains an element close to $\beta^*$. To get some intuition, let us fix some $\beta$ that is close to $\beta^*$ and some $\beta'$ that is far from $\beta$. By the triangle inequality, $\beta'$ therefore should be far from $\beta^*$ as well. As the square loss of a candidate regressor can be viewed as a surrogate for the distance between the regressor and the optimal, it follows that $\sum_{(X,y) \in B} \left(y - X^T\beta\right)^2$ must be significantly less than $\sum_{(X,y) \in B} \left(y - X^T\beta'\right)^2$ over all inlier batches in expectation. Due to $L_2$-$L_4$ hypercontractivity of $X$, we can show that $\left(y - X^T\beta'\right)^2$ must be weakly anti-concentrated. On the other hand, due to the bounds on the higher-order moments of $X$, we can show that $\left(y - X^T\beta\right)^2$ must be sufficiently concentrated around its mean. Combining the two observations then show that Equation (6) must hold with high probability over the inlier distributions. Therefore, setting $\mathcal{W}$ to be the indicator variables for inlier batches must constitute a valid solution to the set of inequalities $\text{IE}(\beta; L, T, R)$ constructed. The full proof can be found in Appendix E.

**Lemma 3.8** (Error Bound). *Let $\alpha \in (0, 1/2)$, $\delta \in (0, 1)$, $n, K \in \mathbb{Z}_+$, $\sigma, R > 0$, and $\beta^* \in \mathbb{R}^d$. Let $L$ be a list of candidate regressors of size $K$, and $\beta \in L$ be a regressor such that $\|\beta - \beta^*\|_2 < R$. Let $n \gg k \ Q^{2/k} \ \alpha^{-2/k}$ be the batch size parameter. Suppose $T$ is a set of $m \gg \min\left(\log(K), d^2\right) \log(1/\delta) \alpha^{-3}$ many batches of size $n$ drawn from the corrupted batch distribution of Definition 1.1. With probability at least $1 - \delta$ over the randomness of $T$, we have that the system $IE(\beta; L, T, R)$ has solutions.*

Proposition 3.6 now follows by an application of the above lemmas.

*Proof of Proposition 3.6.* The pruning procedure proceeds in two steps. First, it filters out the $\beta$ such that $\text{IE}(\beta; L, T, R)$ has no solution. Second, among the remaining regressors, it adds them into the output list as long as it is not $c\left(R + k\alpha^{-1/k}\sigma Q^{1/k}/\sqrt{n}\right)$ -close to some existing regressor in the output list for sufficiently large constant $c$. Then the size of the output list is at most $O(1/\alpha)$ by Lemma 3.7.

It remains to show that there exists some $\beta'$ close to $\beta^*$ in the output list if there exists some $\beta$ satisfying $\|\beta - \beta^*\|_2 < R$ in the input list. By Lemma 3.8, $\beta$ will not be filtered out with probability at least $1 - \delta$ in the first step. In the second step, either $\beta$ is added

to the output list or there must be some $\beta'$ satisfying $\|\beta' - \beta\|_2 \leq O\left(R + k\alpha^{-1/k}\sigma Q^{1/k}/\sqrt{n}\right)$. Hence, we must have $\|\beta' - \beta^*\|_2 \leq O\left(R + k\alpha^{-1/k}\sigma Q^{1/k}/\sqrt{n}\right)$ by the triangle inequality. This concludes the proof. $\square$

### 3.3. Putting Things Together

---
**Algorithm 1** Batch-List-Decode-LRegression (Informal)
---
**Require:** Batch sample access to the linear regression instance, $\sigma, R$ as specified in Theorem 1.3.
1: Initialize a list $L = \{0\}$ to hold candidate regressors.
2: **for** $t = 0, \cdots, \log(R/\sigma)$ **do**
3:      Initialize an empty list $L'$ to store refined candidate regressors.
4:      **for** candidate regressor $\hat{\beta} \in L$ **do**
5:          Take sufficiently many batched samples $T$.
6:          For each $(X, y)$ in $T$, compute the residue $(X, y - \hat{\beta}^\top X)$. Denote the resulting new set of batches as $T'$.
7:          Learn a new list of regressors by running the algorithm from Proposition 3.1 on $T'$, and add the results to $L'$.
8:      **end for**
9:      Replace $L$ with $L'$.
10:     Run algorithm from Proposition 3.6 to prune the list $L$ into one with size $O(1/\alpha)$.
11: **end for**
12: **Return** $L$.
---

In this section, we prove our main theorem, starting with a high-level overview of the algorithm, which mirrors the structure of the robust linear regression algorithm from Diakonikolas et al. (2019) that can tolerate a small constant fraction of outliers. The process begins with estimating $\beta^*$, as outlined in Corollary 3.1, to obtain a list $L$. We then create new linear regression instances by transforming each sample $(X, y)$ into $(X, y - \beta^\top X)$ for every $\beta$ in $L$. This ensures that for at least one transformed instance, the norm of the optimal regressor decreases significantly. Applying Corollary 3.1 to these instances and merging the resulting lists yields a list containing a candidate regressor that is closer to $\beta^*$. We iterate this process until we get a list with an element that is sufficiently close to $\beta^*$. One issue is that the list size will increase exponentially in terms of the number of iterations. To counter this, Proposition 3.6 is employed to prune the list to an optimal size while maintaining the error of the best candidate regressor to within a constant factor. The pseudocode of an informal version of the algorithm is provided in Algorithm 1. See Algorithm 2 in the appendix for the formal version.

*Proof of Theorem 1.3.* Suppose $\|\beta^*\|_2 \leq R$. We claim that with high constant probability the list $L$ will include a can-

didate $\beta$ such that $\|\beta - \beta^*\|_2 \leq O(\sigma)$ after all but the last iteration with respect to $t$. This is trivially true if $R = O(\sigma)$. Otherwise, if $R \gg \sigma$, we use induction to argue that for all $t = 0, \cdots, \max(\log(c_0 R/\sigma), 0) - 1$, with probability at least $(1 - 2\tau)^t$, the list $L$ will contain some candidate regressor $\beta$ such that $\|\beta - \beta^*\|_2 \leq R\,2^{-t}$ after the $t$-th iteration. Note that this implies the above claim for $t = \max(\log(c_0 R/\sigma), 0) - 1$, and for all $t$ considered in the inductive hypothesis we have $\sigma \ll R2^{-t}$.

Conditioned on the existence of such a $\beta$ in the list after the $(t-1)$-th round. By Proposition 3.1, when we execute line 2 in the iteration where $\hat{\beta} = \beta^{(1)}$, we obtain a list of regressors such that with probability at least $1 - \tau$ there exists some $\beta^{(2)} \in L_0$ and some small constant $c$ such that

$$\|\beta^{(2)} + \beta^{(1)} - \beta\|_2$$
$$\leq O\left(\frac{\left(k\,Q^{1/(2k)}\left(R\,2^{-t+1} + \sigma\right)\alpha^{-3/k}\right)}{\sqrt{n}}\right) \leq c\,R\,2^{-t},$$

where the last inequality is true as long as $n \gg k^2\,\alpha^{-6/k}\,Q^{1/k}$, and $R\,2^{-t} \gg \sigma$. The list $L_0$ is now of size $O(\log(1/\tau)/\alpha)$. Thus, after Line 2, by Proposition 3.6, the list $L$ gets pruned into one with size $O(1/\alpha)$ with probability at least $1 - \tau$. Moreover, $L$ still contains some candidate $\beta^{(3)}$ such that

$$\|\beta^{(3)} - \beta\|_2 \leq O\left(c\,R\,2^{-t} + k\alpha^{-1/k}\sigma Q^{1/k}/\sqrt{n}\right) < R\,2^{-t},$$

as long as $c$ is sufficiently small, $n \gg k^2\alpha^{-2/k}Q^{2/k}$, and $R2^{-t} \gg \sigma$. This concludes the induction.

In the last round, since we have $R \leq O(\sigma)$, with a similar argument, we arrive at a list of size $O(1/\alpha)$ that contains some $\beta$ satisfying

$$\|\beta - \beta^*\|_2 \leq O\bigg((k\,Q^{1/(2k)}\,\sigma\,\alpha^{-3/k})/\sqrt{n}$$
$$+ (k\alpha^{-1/k}\sigma Q^{1/k})/\sqrt{n}\bigg) = O\left((k/\sqrt{n})\,Q^{1/k}\,\sigma\,\alpha^{-3/k}\right).$$

It is not hard to see that the total number of samples consumed by the algorithm is at most

$$O\left(\frac{(4kd)^{8k}Q^{-1}}{\alpha} + \frac{\log\left(\log(1/\tau)/\alpha\right)}{\alpha^3}\right)\log(1/\tau)\log(R/\sigma)$$
$$= \tilde{O}\left(\left((4kd)^{8k}Q^{-1}/\alpha + \alpha^{-3}\right)\log(R/\sigma)\right).$$

Moreover, the runtime is polynomial in the sample size times $d^{2k}$, which is the space complexity required for representing a moment $2k$ tensor. $\square$

## Impact Statement

This work is predominantly theoretical and does not present any immediate societal or ethical considerations

requiring special attention. Its contribution lies in advancing foundational understanding rather than influencing direct real-world applications or policy.

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

## Supplementary Material

**Organization**  In Appendix A we discuss work on the problem of mixed linear regression, which is very related to the setting we consider here. In Appendix B we state some basic SoS facts. Then, in Appendix D we adapt proofs from (Diakonikolas et al., 2022a) to show that if the original distribution has certifiably bounded moments, then the uniform distribution over a sufficiently large sample also has certifiably bounded moments. We then use this to prove Lemma 3.4. In Appendix E we prove the lemmas required to get our pruning guarantee in Subsection 3.2. Finally, in Appendix F we present a simple reduction of our problem to the problem of list-decodable linear regression in the non-batch setting.

## A. Related Work on Mixed Linear Regression

**Mixed Linear Regression**  The mixed linear regression setting is when the data is generated by a mixture of $t$ distributions $D_1, \ldots, D_t$, each on $\mathbb{R}^d \times \mathbb{R}$ such that $(X, y) \sim D_i$ is equivalent to $y = \beta_i^T X + \xi$ for $X \sim \mathcal{N}(0, I)$ and $\xi \sim \mathcal{N}(0, \sigma^2)$ (DeVeaux, 1989; Jordan & Jacobs, 1994). We refer the reader to Section 1.2 of (Chen et al., 2020b) for a detailed summary of prior work for this problem. In the non-batch setting, this problem suffers from an exponential dependence on $t$. This is inherent in moment-based approaches, as shown in (Chen et al., 2020b). The most efficient algorithm for the problem is due to (Diakonikolas & Kane, 2020) which runs in time and needs samples quasi-polynomial in $t$.

In the batch setting, this was first studied for covariates drawn from $\mathcal{N}(0, I)$ by (Kong et al., 2020a;b). Here, all the samples from each batch belong to a single component. (Kong et al., 2020b) design an algorithm that requires $O(d)$ batches of size $O(\sqrt{t})$ to solve the problem efficiently (including in terms of the parameter $t$). Subsequently, (Kong et al., 2020a) uses the sum-of-squares hierarchy to design a class of algorithms that can trade between the batch size and sample complexity while being robust to a small fraction of outliers. [4] Finally, (Jain et al., 2023) greatly generalize the scope by designing an algorithm that can recover the regressors for all components such that at least an $\alpha$ fraction of the batches satisfy a linear-regression model with variance in the noise bounded by $\sigma^2$. Their algorithm works even when the covariates for each component are different, varying, and heavy-tailed. They do this by allowing for batches of nonuniform size. They require $\tilde{O}(d/\alpha^2)$ batches of size $\geq 2$ and $\tilde{\Omega} \min(\sqrt{t}, 1/\sqrt{\alpha})/\alpha$ batches of size $\tilde{\Omega} \min(\sqrt{t}, 1/\sqrt{\alpha})$. This is very close to the list-decodable setting we study in this paper; however, we do not allow for nonuniform batch sizes. Even so, our algorithm can improve the batch size required by a constant power of the algorithm designer's choice in the exponent.

## B. Further background of SoS Proofs and Moment Bounds

It is a standard fact that several commonly used inequalities like the triangle inequality, Cauchy-Schwartz, or AM-GM inequalities have an SoS version.

**Fact B.1** (SoS Cauchy-Schwartz and Hölder (see, e.g., (Hopkins, 2018))). *Let $f_1, g_1, \ldots, f_n, g_n$ be indeterminates. Then,*

$$\frac{\big|}{2}^{f_1,\ldots,f_n,g_1,\ldots,g_n} \left\{ \left( \frac{1}{n} \sum_{i=1}^n f_i g_i \right)^2 \leq \left( \frac{1}{n} \sum_{i=1}^n f_i^2 \right) \left( \frac{1}{n} \sum_{i=1}^n g_i^2 \right) \right\} .$$

**Fact B.2** (SoS Triangle Inequality). *If $k$ is an even integer, $\frac{\big|}{k}^{a_1, a_2, \ldots, a_n} \left\{ \left( \sum_{i=1}^n a_i \right)^k \leq n^k \left( \sum_{i=1}^n a_i^k \right) \right\}$.*

**Fact B.3** (SoS AM-GM Inequality, see, e.g., Chapter 2 of (Hardy et al., 1952)). *Let $k$ be an even integer, and $\{w_i\}_{i=1}^n$ be integers such that $\sum_{i=1}^n w_i = k$. Then it holds that*

$$\frac{\big|}{k}^{x_1 \ldots, x_n} \left\{ \prod_i x_i^{w_i} \leq \sum_{i=1}^n \frac{w_i}{k} x_i^k \right\} .$$

Using these inequalities, we can construct SoS proofs for bounds of moments of sum of i.i.d. random variables with SoS certifiably bounded moments. The non-sos version of the inequality is commonly known as the Marcinkiewicz-Zygmund inequality.

*Proof of Lemma 3.3.*  For notational convenience, we define $y_i = p(v, X_i) - \mathbf{E}[p(v, X_i)]$. Note that each $y_i$ is a degree-$t$ polynomial in $v$ and $X_i$. If we expand $\left( \sum_{i=1}^n y_i \right)^k$, we get $n^k$ many monomials of the form $\prod_{j=1}^k y_{\sigma_j}$ for some $\sigma_j \in [n]^k$.

---

[4]This is similar in flavor to what we do in this paper but for the much harder problem of list-decodable linear regression.

If the degree of some $y_i$ is 1, the expected value of that monomial will be 0 since $\mathbf{E}[y_i] = \mathbf{E}[p(v, X_i) - \mathbf{E}[p(v, X_i)]] = 0$. Hence, $\mathbf{E}[\prod_{j=1}^{k} y_{\sigma_j}]$ is non-zero only if the number of variables appeared is at most $k/2$ since otherwise some $y_i$ must have degree-1 by the pigeonhole principle. By a simple counting argument, we have that the number of monomials with non-zero expectations is then at most $\binom{n}{(k/2)} k^{k/2}$. Let $\prod_{i=1}^{n} y_i^{w_i}$ be one of such monomial with non-zero expectation, where $\sum_{i=1}^{n} w_i = k$. We can bound its expectation from above by

$$\{\|v\|_2^2 = 1\} \left|\frac{2kt}{v}\right. \mathbf{E}\left[\prod_{i=1}^{n} y_i^{w_i}\right] \leq \sum_{i=1}^{n} \frac{w_i}{k} \mathbf{E}\left[y_i^k\right] \leq M,$$

where the first inequality is by Fact B.3, and the second inequality is by our assumption that $\{\|v\|_2^2 = 1\} \left|\frac{2kt}{v}\right. \mathbf{E}[y_i^k] = \mathbf{E}\left[(p(v, X_i) - \mathbf{E}[p(v, X_i)])^k\right] \leq M$. Since there are at most $n^{k/2} k^{k/2}$ such monomials with non-zero expectation, it then follows that

$$\{\|v\|_2^2 = 1\} \left|\frac{2kt}{v}\right. \mathbf{E}\left[\left(\sum_{i=1}^{n} y_i\right)^k\right] \leq (kn)^{k/2} M.$$

$\square$

# C. Algorithm Pseudocode

---
**Algorithm 2** Batch-List-Decode-LRegression
---
**Require:** Batch sample access to the linear regression instance, and $\alpha, \sigma, R, k$ as specified in Theorem 1.3.
1: Initialize $L = \{0\}$.
2: Set failure probability $\tau = 0.001/\log(R/\sigma)$.
3: Let $c_0$ be some sufficiently small constant and $C$ be some sufficiently large constant.
4: **for** $t = 0, \cdots, \max(\log(c_0 R/\sigma), 0)$ **do**
5:     Initialize $L_0 = \{0\}$
6:     **for** candidate regressor $\hat{\beta} \in L$ **do**
7:         **for** $r = 0, \cdots, \log(1/\tau)$ **do**
8:             Take a batch of $C (2dk)^{10k}/\alpha$ samples $T$.
9:             For each $(X, y)$, compute $(X, y - \hat{\beta}^\top X)$. Denote the new set of batches as $T'$.
10:            Learn a list $L_1$ of regressors by running the algorithm from Proposition 3.1 on $T'$.
11:            Add the candidate regressors $\{\hat{\beta}' + \hat{\beta} \mid \hat{\beta}' \in L_1\}$ into $L_0$.
12:         **end for**
13:     **end for**
14:     Set $L = L_0$.
15:     Draw $C \min\left(\log(\log(1/\tau)/\alpha), d^2\right) \log(1/\tau)\alpha^{-3}$ batch of samples $T'$.
16:     Run algorithm from Proposition 3.6 on $T'$ to prune the list $L$ with failure probability $\tau$.
17: **end for**
18: **Return** $L$.
---

# D. Certifiably Bounded Moments of the Regressor Estimator

In this subsection, we give the proof of Lemma D.4. We first give several preliminary lemmas regarding the concentration properties of empirical higher order moment tensors of distribution with bounded central moments. The proof is similar, for example, to Lemma A.4 from (Diakonikolas et al., 2022b).

**Lemma D.1.** *Let $D$ be a distribution over $\mathbb{R}^d$ with mean $\mu$ and $t \in \mathbb{Z}_+$. Suppose that $D$ has its covariance bounded from above by $\kappa I$, and its degree-$2t$ central moments bounded by $F$, i.e., $\mathbf{E}_{X \sim D}\left[\left|\mathbf{v}^T(X - \mu)\right|^{2t}\right] \leq F$. Let $X_1, \ldots, X_m$ be $m$ i.i.d. samples from $D$. The following inequalities hold with high constant probability.*

$$\left\|\mathop{\mathbf{E}}_{i \sim [m]}[(X_i - \mu)^{\otimes t}] - \mathop{\mathbf{E}}_{X \sim D}[(X - \mu)^{\otimes t}]\right\|_\infty \leq O\left(d^t \sqrt{tF/m}\right).$$

*Define $\overline{\mu} = \frac{1}{m} \sum_{i=1}^{m} X_i$. We also have that*

$$\|\mu - \overline{\mu}\|_2 \leq O\left(\sqrt{\kappa d/m}\right).$$

*Proof.* Note that each entry within the tensor $\mathbf{E}_{X \sim D}\left[(X - \mu)^{\otimes t}\right]$ is of the form $\mathbf{E}_{X \sim D}\left[T(x - \mu)\right]$, where $T : \mathbb{R}^d \mapsto \mathbb{R}$ is some monomial of degree $t$. Fix some degree $t$ monomial $T : \mathbb{R}^d \mapsto \mathbb{R}$, and consider the random variable $Y = T(X - \mu)$, where $X \sim D$. The corresponding entry within the tensor $\mathbf{E}_{i \sim [m]}\left[(X_i - \mu)^{\otimes t}\right]$ has the same distribution as the average of $m$ i.i.d. copies of $Y$. We will bound from above the variance of $Y$. We will need the following claim regarding expectations of monomials.

**Claim D.2.** *Let $t \in Z_+$ be an even integer. Suppose the distribution $D$ has its $t$-th central moments bounded from above by $M$. Let $T : \mathbb{R}^d \mapsto \mathbb{R}$ be a monomial of degree $t$. Then it holds*

$$\mathbf{E}_{X \sim D}\left[T(X - \mu)\right] \leq tM.$$

*Proof.* Suppose $T(X - \mu) = \prod_{i=1}^{d}(X_i - \mu_i)^{s_i}$, where $\sum_{i=1}^{d} s_i = t$. Then we have

$$
\begin{aligned}
\mathbf{E}_{X \sim D}\left[T(X - \mu)\right] &\leq \mathbf{E}_{X \sim D}\left[\left(\max_{i \in [d]:s_i > 0} |X_i - \mu_i|\right)^t\right] \\
&= \mathbf{E}_{X \sim D}\left[\max_{i \in [d]:s_i > 0}\left(|X_i - \mu_i|^t\right)\right] \\
&\leq \sum_{i \in [d]:s_i > 0} \mathbf{E}_{X \sim D}\left[(X_i - \mu_i)^t\right] \leq tM,
\end{aligned}
$$

where in the first inequality we bound $(X_i - \mu_i)$ from above by $\max_{i \in [d]:s_i > 0} |X_i - \mu_i|$, in the second inequality we bound the maximum of a set of non-negative numbers by their sum, and in the last inequality we use the fact that there are at most $t$ non-zero $s_i$'s, and that $D$ has its $t$-th central moments bounded from above by $M$. This concludes the proof of Claim D.2. $\square$

We can therefore bound from above the variance of $Y$ by

$$\mathbf{Var}[Y] \leq \mathbf{E}[Y^2] = \mathbf{E}[T^2(X - \mu)] \leq O\left(tF\right),$$

where in the last inequality we note that $T^2$ is a degree $2t$ monomial, and thus we can apply Claim D.2. Hence, by Chebyshev's inequality, we have that

$$\left|\mathbf{E}_{i \sim [m]}\left[T(X_i - \mu)\right] - \mathbf{E}_{X \sim D}\left[T(X - \mu)\right]\right| \leq O\left(d^t \sqrt{tF/m}\right),$$

with probability at least $1 - o\left(d^{-t}\right)$. It then follows from the union bound that

$$\left\|\mathbf{E}_{i \sim [m]}[(X_i - \overline{\mu})^{\otimes t}] - \mathbf{E}_{X \sim D}[(X - \mu)^{\otimes t}]\right\|_\infty \leq O\left(d^t \sqrt{tF/m}\right). \tag{7}$$

with high constant probability. Lastly, we bound from above $\|\mu - \overline{\mu}\|_2$. Since $D$ has its covariance bounded from above by $\kappa I$, it holds that the random vector $\mu - \overline{\mu}$ has mean $0$ and covariance bounded from above by $\kappa/mI$. Hence, the expected squared $\ell_2$ norm of the vector is at most $\kappa d/m$. It then follows from Markov's inequality that $\|\mu - \overline{\mu}\|_2 \leq O\left(\sqrt{\kappa d/m}\right)$ holds with high constant probability. This concludes the proof of Lemma D.1. $\square$

The next lemma provides a sum of square proof that bounds from above the square of a polynomial in terms of its coefficients.

**Lemma D.3.** *Let $p(v) = v^{\otimes t} A v^{\otimes t}$ for some $d^t \times d^t$ matrix $A$ with $\|A\|_\infty \leq a$. Then*

$$\left|\frac{v}{2t}\right. p(v) \leq ad^t \|v\|_2^{2t}.$$

*Proof.* Since the Frobenious norm of $A$ is at most $ad^t$, we have that $A$ is bounded from above by $ad^t I$ in Lowner order. Thus, we can write $v^{\otimes t}(ad^t I)v^{\otimes t} - v^{\otimes t}Av^{\otimes A}$ as a sum of squares by diagonalizing $I$ and $A$. The lemma then follows by noting that the expression is exactly $ad^t \|v\|_2^{2t} - p(v)$. $\square$

We can now put these together to get the lemma we need.

**Lemma D.4.** *Let $D$ be a distribution over $\mathbb{R}^d$ with mean $\mu$ and $t$ be a positive even integer. Assume that (i) the covariance of $D$ is bounded from above by $\kappa I$, (ii) the degree-$2t$ central moments of $D$ is bounded from above by $F > 0$, and (iii) there exists $M > 0$ such that $D$ has $(M, t, K)$-certifiably bounded moments. Let $S = \{X_1, \ldots, X_m\}$ be a set of $m$ i.i.d. samples from $D$, $D'$ be the uniform distribution over $S$, and $\overline{\mu} := \mathbf{E}_{X \sim D'}[X]$. If $m \gg (td)^{4t}(F/M^2) + d\kappa M^{-2/t}$, then $D'$ will have $(2^{t+2}M, t, K)$-certifiably bounded moments with probability at least $0.9$.*

*Proof.* From Lemma D.1 and that $m \gg (td)^{4t}(F/M^2) + d\kappa M^{-2/t}$, we have that the $\ell_\infty$ norm of the difference between the expected and empirical $t$-th tensors $(X - \mu)^{\otimes t}$ of $D$ and $D'$ is small, i.e.,

$$\left\| \mathop{\mathbf{E}}_{i \sim [m]}[(X_i - \mu)^{\otimes t}] - \mathop{\mathbf{E}}_{X \sim D}[(X - \mu)^{\otimes t}] \right\|_\infty \leq \frac{M}{\sqrt{d^t}} , \tag{8}$$

and that the empirical mean and the distribution mean are close, i.e.,

$$\|\mu - \overline{\mu}\|_2 \leq M^{1/t} \tag{9}$$

with high constant probability.

Let $q(v) := \mathbf{E}_{i \sim [m]}[\langle v, X_i - \mu \rangle^t] - \mathbf{E}_{X \sim D}[\langle v, X - \mu \rangle^t]$. Combining Lemma D.3 and Equation (8) gives that

$$\left|_{\frac{v}{t}} \mathop{\mathbf{E}}_{i \sim [m]} \left[\langle v, X_i - \mu \rangle^t\right] - \mathop{\mathbf{E}}_{X \sim D}\left[\langle v, X - \mu \rangle^t\right]\right.$$

$$\leq \sqrt{d^t} \|v\|_2^t \left\| \mathop{\mathbf{E}}_{i \sim [m]}[(X_i - \mu)^{\otimes t}] - \mathop{\mathbf{E}}_{X \sim D}[(X - \mu)^{\otimes t}] \right\|_\infty \leq \|v\|_2^t M. \tag{10}$$

Observe that

$$\left|_{\frac{v}{t}} \mathop{\mathbf{E}}_{i \sim [m]} \left[\langle v, X_i - \mu \rangle^t\right] = \mathop{\mathbf{E}}_{i \sim [m]} \left[\langle v, X_i - \mu \rangle^t\right] - \mathop{\mathbf{E}}_{X \sim D}\left[\langle v, X - \mu \rangle^t\right] + \mathop{\mathbf{E}}_{X \sim D}\left[\langle v, X - \mu \rangle^t\right]\right.$$

$$\leq 2\|v\|_2^t M , \tag{11}$$

where in the second line we use Equation (10) and our assumption that $D$ has certifiably bounded central moments.

Lastly, to prove bounded central moments of $D'$ (the uniform distribution over the samples in $S$), we note that

$$\left|_{\frac{v}{t}} \mathop{\mathbf{E}}_{i \sim [m]} \left[\langle v, X_i - \overline{\mu} \rangle^t\right] \leq 2^t \mathop{\mathbf{E}}_{i \sim [m]} \left[\langle v, X_i - \mu \rangle^t\right] + 2^t \mathop{\mathbf{E}}_{i \sim [m]} \left[\langle v, \mu - \overline{\mu} \rangle^t\right]\right.$$

$$\leq 2^{t+1} \mathop{\mathbf{E}}_{i \sim [m]} \left[\langle v, X_i - \overline{\mu} \rangle^t\right] + 2^t \|v\|_2^t \|\mu - \overline{\mu}\|_2^t$$

$$\leq 2^{t+2} \|v\|_2^t M ,$$

where in the first line we use the SoS triangle inequality (Fact B.2), in the second line we use SoS Cauchy's inequality (Fact B.1), and the last inequality follows from Equations (9) and (11). $\square$

**Lemma D.5** (SoS Moment Bound). *Let $\alpha \in (0, 1/2)$, $\sigma > 0$, $k \in \mathbb{Z}^+$, $\beta^* \in \mathbb{R}^d$. Let $T$ be a set of $m$ batches drawn according to the distribution $D_{\beta^*}$ defined in Definition 1.1, and batch size $n$. Assume that the clean covariates distribution $X$ satisfies Assumption 1.2 and $k \leq \Delta/2$. Define $Z_B = \frac{1}{n} \sum_{(X,y) \in B} Xy$. Suppose $m \gg \left((4kd)^{8k}Q^{-1} + 1\right)\alpha^{-1}$. Then the following holds with probability at least $0.9$: (a) $\{Z_B \mid B \in T\}$ has $(M, 2k, 4k)$-certifiably bounded moments for some $M = O((2k)^{2k}/n^k) Q \left(\sigma^{2k} + 2\|\beta^*\|_2^{2k}\right)$, and (b) $\mathbf{Cov}_{B \sim T}[Z_B] \preceq O((\|\beta^*\|_2^2 + \sigma^2)/n)\mathbf{I}$.*

*Proof.* We first prove that the population version of the above inequality has SoS proof. Specifically, we show that

$$\{\|v\|_2^2 = 1\} \left|_{\frac{v}{4k}} \mathop{\mathbf{E}}_{B \sim D_{\beta^*}} \left[\left(v^\top \left(Z_B - \mathop{\mathbf{E}}_{B' \sim T}[Z_{B'}]\right)\right)^{2k}\right] \leq \frac{(2k)^{2k}}{n^k} Q \left(\sigma^{2k} + \|\beta^*\|_2^{2k}\right). \tag{12}$$

We can rewrite the left hand side as

$$
\operatorname*{\mathbf{E}}_{(X_i,y_i)\sim P_{\beta^*}\forall i\in[n]}\left[\left(\left(\frac{1}{n}\sum_{i=1}^n v^\top X_i y_i - \operatorname*{\mathbf{E}}_{X,y\sim P_{\beta^*}}[v^\top Xy]\right)\right)^{2k}\right]
$$
$$
= \frac{1}{n^{2k}}\operatorname*{\mathbf{E}}_{(X_i,y_i)\sim P_{\beta^*}\forall i\in[n]}\left[\left(\sum_{i=1}^n\left(v^\top X_i y_i - v^\top\beta^*\right)\right)^{2k}\right]. \tag{13}
$$

We first show that $\mathbf{E}\left[\left(v^\top(Xy-\beta^*)\right)^{2k}\right]$ is SoS-certifiably bounded. In particular, we claim that

$$
\{\|v\|_2^2 = 1\}\Big|_{4k}^{v}\operatorname*{\mathbf{E}}_{(X,y)\sim P_{\beta^*}}\left[\left(v^\top(Xy-\beta^*)\right)^{2k}\right] \le (2k)^k\,Q\,\left(\sigma^{2k}+\|\beta^*\|_2^{2k}\right). \tag{14}
$$

Note that

$$
\{\|v\|_2^2 = 1\}\Big|_{4k}^{v}\operatorname*{\mathbf{E}}_{(X,y)\sim P_{\beta^*}}\left[\left(v^\top(Xy-\beta^*)\right)^{2k}\right] = \operatorname*{\mathbf{E}}_{(X,y)\sim P_{\beta^*}}\left[\left(v^\top XX^\top\beta^* + v^\top X\xi - v^\top\beta^*\right)^{2k}\right]
$$

$$
\le 3^{2k}\operatorname*{\mathbf{E}}_{(X,y)\sim P_{\beta^*}}\left[\left(v^\top XX^\top\beta^*\right)^{2k} + \left(v^\top X\xi\right)^{2k} + \left(v^\top\beta^*\right)^{2k}\right],
$$

where in the last line we apply the SoS triangle inequality (Fact B.2). We then tackle the three terms separately. For the first term, we note that

$$
\{\|v\|_2^2 = 1\}\Big|_{4k}^{v}\operatorname*{\mathbf{E}}_{(X,y)\sim P_{\beta^*}}\left[\left(v^\top XX^\top\beta^*\right)^{2k}\right] \le \frac{\|\beta^*\|_2^{2k}}{2}\operatorname*{\mathbf{E}}_{(X,y)\sim P_{\beta^*}}\left[(v^\top X)^{4k} + (X^\top\beta^*/\|\beta^*\|_2)^{4k}\right]
$$
$$
\le \|\beta^*\|_2^{2k}Q.
$$

where in the first inequality we use the SoS AM-GM inequality (Fact B.3), and in the second inequality we use the assumption that the degree-$4k$ moments of $X$ are SoS certifiably bounded by $Q$ (Assumption 1.2). For the second term, note that

$$
\{\|v\|_2^2 = 1\}\Big|_{2k}^{v}\operatorname*{\mathbf{E}}_{(X,y)\sim P_{\beta^*}}\left[\left(v^\top X\xi\right)^{2k}\right] = \operatorname*{\mathbf{E}}_{(X,y)\sim P_{\beta^*}}\left[\left(v^\top X\right)^{2k}\right]\mathbf{E}[\xi^{2k}] \le (2k)^k\sigma^{2k}Q,
$$

where in the first equality we use that $X$ and $\xi$ are independent, and in the second inequality we use again the assumption on the moments of $X$ and that the degree $2k$ moments of $\xi$ is bounded by $(2k)^k\sigma^{2k}$. For the last term, we note that $(v^\top\beta^*)^{2k} \le \|v\|_2^{2k}\|\beta^*\|_2^{2k}$ by an application of the SoS Cauchy's inequality (Fact B.1). Combining the above analysis then shows Equation (14).

By Lemma 3.3, we then have the SoS proof

$$
\{\|v\|_2^2 = 1\}\Big|_{2k}^{v}\operatorname*{\mathbf{E}}_{(X_i,y_i)\sim P_{\beta^*}\forall i\in[n]}\left[\left(\sum_{i=1}^n\left(v^\top X_i y_i - v^\top\beta^*\right)\right)^{2k}\right] \le n^k(2k)^{2k}Q\left(\sigma^{2k}+\|\beta^*\|_2^{2k}\right) \tag{15}
$$

Combining this with Equation (13) then yields an SoS proof for Equation (12).

In order to establish an SoS proof for the empirical moments, we will additionally need to bound the covariance of the empirical distribution over $\{Z_B\}_{B\in T}$. Since an SoS proof on the bound of the covariance is not needed, we can readily apply the $L_2 - L_4$ hypercontractivity of $X$. In particular, this shows that $\mathbf{E}[(u^TX)^4] \le O(1)\left(\mathbf{E}\left[(u^TX)^2\right]\right)^2 \le O(1)$ for any unit vector $u$. With an argument almost identical to the SoS bound on the degree-$2k$ moments, we can show that

$$
\operatorname*{\mathbf{E}}_{(X_i,y_i)\sim P_{\beta^*}\forall i\in[n]}\left[\left(\frac{1}{n}\sum_{i=1}^n\left(v^\top X_i y_i - v^\top\beta^*\right)\right)^2\right] \le O\left(\frac{\sigma^2+\|\beta^*\|_2^2}{n}\right).
$$

This shows property (b) in the lemma.

Let $C$ be a sufficiently large constant. The SoS proof for the empirical moments then follows by an application of Lemma D.4 with $t = 2k$, $\kappa := C\frac{1}{n} \left(\sigma^2 + \|\beta^*\|_2^2\right)$, $M := \frac{(2k)^{2k}}{n^k} Q \left(\sigma^{2k} + \|\beta^*\|_2^{2k}\right)$, $F := \frac{(4k)^{4k}}{n^{2k}} Q \left(\sigma^{4k} + \|\beta^*\|_2^{4k}\right) \leq 2^{4k} M^2 Q^{-1}$, and

$$m \gg (2kd)^{8k} \, 2^{4k} \, Q^{-1} + \frac{d}{n} \left(\sigma^2 + \|\beta^*\|_2^2\right) \, M^{-1/k} + 1.$$

It is not hard to see that

$$(2kd)^{8k} \, 2^{4k} \, Q^{-1} + \frac{d}{n} \left(\sigma^2 + \|\beta^*\|_2^2\right) \, M^{-1/k} + 1 \leq O(1) \left((4kd)^{8k} Q^{-1} + dQ^{-1/k} + 1\right) \leq O\left((4kd)^{8k} Q^{-1}\right),$$

where the last inequality can be shown by examining the cases where $dQ^{-1/k} \geq 1$ and $dQ^{-1/k} < 1$ separately. This concludes the proof of Lemma 3.4. $\qquad\square$

# E. Pruning Procedure and its Analysis

The main theorem for this subsection is the following:

**Proposition E.1** (Pruning Lemma). *Let $\alpha \in (0, 1/2)$, $\delta \in (0, 1)$, $k, n \in \mathbb{Z}_+$, $\sigma, R > 0$, and $\beta^* \in \mathbb{R}^d$. Let $L \subset \mathbb{R}^d$ be a list of candidate regressors, and $\beta \in L$ be a regressor such that $\|\beta - \beta^*\|_2 < R$. Assume that the batch size $n$ satisfies that $n \gg k \, Q^{2/k} \, \alpha^{-2/k}$ and $k \leq \Delta/2$. Then there exists an algorithm Pruning that takes the list $L$, and the numbers $\alpha, \delta, R$ as input, draws $m = O\left(\min\left(\log(|L|), d^2\right) \, \log(1/\delta) \, \alpha^{-3}\right)$ many batches from the corrupted batch distribution of Definition 1.1, runs in time $\mathrm{poly}(dm|L|)$, and outputs at most $O(1/\alpha)$ candidate regressors $L' \subseteq L$ such that there is at least one regressor $\beta \in L'$ satisfying $\|\beta - \beta^*\|_2^2 \leq O\left(R + k\alpha^{-1/k}\sigma Q^{1/k}/\sqrt{n}\right)$ with probability at least $1 - \delta$ over the randomness of the batches drawn.*

The Pruning algorithm involves two phases: initially, it filters regressors $\beta \in L$ by retaining those matching a certain set of solvable linear inequalities. Then, it selects a subset of the remaining regressors, ensuring each pair is adequately distant. Lemmas 3.7 and 3.8 respectively prove that the refined list is not excessively large and contains a regressor near the optimal $\beta^*$, given one exists in the original list $L$. The proof of Proposition 3.6 follows from the above two lemmas.

For each regressor, we restate the set of linear inequalities $\mathrm{IE}(\beta; L, T, R)$ in the weighting function $\mathcal{W}$ over the set of batches $T$.

$$\sum_{B \in T} \mathcal{W}(B) \geq 0.9\alpha|T|, \tag{16}$$

$\forall \beta' \in L$ such that $\|\beta' - \beta\| \geq c\left(R + k\alpha^{-1/k}\sigma Q^{1/k}/\sqrt{n}\right)$ for some sufficiently large constant $c$,

$$\sum_{B \in T} \mathbb{1}\left\{\sum_{(X,y) \in B} \left(y - X^\top \beta\right)^2 \leq \sum_{(X,y) \in B} \left(y - X^\top \beta'\right)^2\right\} \mathcal{W}(B) \leq \frac{\alpha}{20} \sum_{B \in T} \mathcal{W}(B). \tag{17}$$

We now show there cannot be too many regressors whose associated linear inequalities are satisfiable subject to the constraint that they are all sufficiently separated. This mainly comes from the observation that Condition 17 enforces the soft clusters associated with two sufficiently separated candidate regressors must have small intersection.

**Lemma E.2** (List Size Bound). *Let $R > 0$, and $L$ be a list of candidate regressors. Let $T$ be a set of batches. Let $L' \subseteq L$ be a sublist of candidate regressors satisfying the following conditions: (1) $\mathrm{IE}(\beta; L, T, R)$ has solutions for each $\beta \in L'$, and (2) $\|\beta_1 - \beta_2\|_2 \geq c\left(R + k\alpha^{-1/k}\sigma Q^{1/k}/\sqrt{n}\right)$ for any two $\beta_1, \beta_2 \in L'$. Then it holds the size of $L'$ is at most $O(1/\alpha)$.*

*Proof.* Let $I$ be a set of weighting functions $\mathcal{W} : T \mapsto [0, 1]$ over batches. We first define the union and disjoint operators for weighting functions as follows

$$\left(\bigcup_{\mathcal{W} \in I} \mathcal{W}\right)(B) = \max_{\mathcal{W} \in I} \mathcal{W}(B), \quad \left(\bigcap_{\mathcal{W} \in I} \mathcal{W}\right)(B) = \min_{\mathcal{W} \in I} \mathcal{W}(B).$$

Moreover, for a weighting function $\mathcal{W} : T \mapsto [0, 1]$, we define $\mathcal{W}(T) = \sum_{B \in T} \mathcal{W}(B)$. Let $\beta_1, \beta_2$ be two vectors from the sublist $L'$, and $\mathcal{W}_1, \mathcal{W}_2$ be the solutions of $\text{IE}(\beta_1; L, T, R)$ and $\text{IE}(\beta_2; L, T, R)$ respectively. We proceed to argue that $(\mathcal{W}_1 \cap \mathcal{W}_2)(T) < 0.1\alpha \left( \mathcal{W}_1(T) + \mathcal{W}_2(T) \right)$. For the sake of contradiction, we assume that

$$(\mathcal{W}_1 \cap \mathcal{W}_2)(T) > 0.1\alpha \left( \mathcal{W}_1(T) + \mathcal{W}_2(T) \right). \tag{18}$$

Define the following two subsets of batches:

$$\mathcal{E}_1 := \left\{ B \in T : \sum_{(X,y) \in B} \left( y - X^\top \beta_1 \right)^2 \leq \sum_{(X,y) \in B} \left( y - X^\top \beta_2 \right)^2 \right\},$$

$$\text{and } \mathcal{E}_2 := \left\{ B \in T : \sum_{(X,y) \in B} \left( y - X^\top \beta_2 \right)^2 \leq \sum_{(X,y) \in B} \left( y - X^\top \beta_1 \right)^2 \right\}.$$

Since each batch $B$ belongs to either $\mathcal{E}_1$ or $\mathcal{E}_2$, we have either $(\mathcal{W}_1 \cap \mathcal{W}_2)(\mathcal{E}_1) \geq (\mathcal{W}_1 \cap \mathcal{W}_2)(T)/2$ or $(\mathcal{W}_1 \cap \mathcal{W}_2)(\mathcal{E}_2) \geq (\mathcal{W}_1 \cap \mathcal{W}_2)(T)/2$. Without loss of generality, assume that we are in the former case. This then implies that

$$\sum_{B \in T} \mathbb{1} \left\{ \sum_{(X,y) \in B} \left( y - X^\top \beta_1 \right)^2 \leq \sum_{(X,y) \in B} \left( y - X^\top \beta_2 \right)^2 \right\} \mathcal{W}_1(B)$$

$$\geq 0.05\alpha \left( \mathcal{W}_1(T) + \mathcal{W}_2(T) \right) > \frac{\alpha}{20} \sum_{B \in T} \mathcal{W}_1(T),$$

which contradicts Equation (17) for $\beta_1$. This shows the opposite of Equation (18).

Lastly, assume that there are more than $4/\alpha$ many candidate regressors in the sublist $L'$ for the sake of contradiction. Arbitrarily pick $\ell = \lceil 4/\alpha \rceil$ many regressors from $L'$, and let $\mathcal{W}_1, \ldots, \mathcal{W}_\ell$ be the solutions to the linear inequalities associated with the candidate regressors picked. Then,

$$\begin{aligned}
|T| &\geq \left( \bigcup_{i=1}^\ell \mathcal{W}_i \right)(T) \\
&\geq \sum_{i=1}^\ell \mathcal{W}_i(T) - \sum_{i < j \in [\ell]} (\mathcal{W}_i \cap \mathcal{W}_j)(T) \\
&\geq \sum_{i=1}^\ell \mathcal{W}_i(T) - 0.1\alpha \sum_{i < j \in [\ell]} (\mathcal{W}_i(T) + \mathcal{W}_j(T)) \\
&= (1 - 0.1\alpha(\ell - 1)) \sum_{i=1}^\ell \mathcal{W}_i(T) \\
&\geq (1 - 0.1(\ell - 1)\alpha)\, \ell(0.9\alpha)|T| \\
&\geq 2.88|T|,
\end{aligned}$$

where in the first line we use the fact that the weights are bounded from above by 1, in the second line we use the approximate inclusion-exclusion principle, in the third line we use the opposite of Equation (18), in the fourth line we use the elementary fact that $\sum_{i \neq j \in [\ell]} (x_i + x_j) = (\ell - 1) \sum_{i=1}^\ell x_i$, in the fifth line we use $\mathcal{W}_i(T) \geq 0.9\alpha|T|$ as they need to satisfy Condition 5, and in the last line we use the definition of $\ell = \lceil 4/\alpha \rceil$. This is clearly a contradiction, and hence concludes the proof of Lemma 3.7. $\qquad\square$

Next we show that the set of linear inequalities constructed for some $\beta$ admit solutions with high probability as long as $\beta$ is close to $\beta^*$.

**Lemma E.3** (Error Bound). *Let $\alpha \in (0, 1/2)$, $\delta \in (0, 1)$, $n, K \in \mathbb{Z}_+$, $\sigma, R > 0$, and $\beta^* \in \mathbb{R}^d$. Let $L$ be a list of candidate regressors of size $K$, and $\beta \in L$ be a regressor such that $\|\beta - \beta^*\|_2 < R$. Let $n \gg k\, Q^{2/k}\, \alpha^{-2/k}$ be the batch size parameter. Suppose $T$ is a set of $m \gg \min\left( \log(K), d^2 \right)\, \log(1/\delta)\, \alpha^{-3}$ many batches of size $n$ drawn from the corrupted batch distribution of Definition 1.1. With probability at least $1 - \delta$ over the randomness of $T$, we have that the system $\text{IE}(\beta; L, T, R)$ has solutions.*

To prove Lemma 3.8, we will make essential use of the following anti-concentration inequalities.

**Fact E.4** (Paley–Zygmund Inequality). *If $Z \geq 0$ is a positive random variable with finite variance, and $\theta \in [0, 1]$, then it holds*

$$\Pr[Z \geq \theta \mathbf{E}[Z]] \geq (1 - \theta)^2 \frac{\mathbf{E}[Z]^2}{\mathbf{E}[Z^2]}.$$

Combining the above with our distributional assumption that the clean covariates distribution satisfies $L_2$-$L_4$ hypercontractivity, we obtain the following *weak anti-concentration* property.

**Corollary E.5** (Weak Anti-concentration). *Let $v$ be a unit vector in $\mathbb{R}^d$, and $X$ be a random unit vector satisfying Assumption 1.2. Then it holds*

$$\Pr[(vX)^2 \geq 1/2] \geq \Omega(1).$$

We are now ready to give the proof of Lemma 3.8.

*Proof of Lemma 3.8.* Let $\beta$ be a regressor within the list such that $\|\beta - \beta^*\|_2 < R$. Our goal is to show that the associated linear inequalities $IE(\beta; L, T, R)$ admits solutions. In particular, we claim that setting $\mathcal{W}(B) = 1$ for all inlier batch $B$ and $\mathcal{W}(B) = 0$ for all outlier batch $B$ gives a solution. Condition 5 is satisfied since in expectation there should be $\alpha$-fraction of inlier batches. Since we take $C \log(\delta/\alpha)/\alpha^2$ many batches, the actual fraction of inlier batches should be at least $0.9\alpha$ with probability at least $1 - \delta$ when $C$ is sufficiently large by the Chernoff bound.

Next we show Condition 6 is satisfied with high probability over the randomness of $T$. Fix some $\beta'$ satisfying $\|\beta' - \beta\|_2 \gg R + k\alpha^{-1/k}\sigma Q^{1/k}/\sqrt{n}$. We will analyze the random variable

$$Z_{\beta'}(B) := \sum_{(X,y) \sim B} \left(y - X^\top \beta'\right)^2 - \sum_{(X,y) \sim B} \left(y - X^\top \beta\right)^2,$$

where $B \sim D_{\beta^*}$. Recall that we have $y = X^\top \beta^* + \xi$, where $\xi \sim \mathcal{N}(0, \sigma^2)$. We will rewrite $Z(\beta')$ slightly with the random variables $\{(X^{(i)}, \xi^{(i)})\}_{i=1}^n$, where each $X^{(i)}$ is drawn independently from a distribution satisfying Assumption 1.2 , and each $\xi^{(i)}$ is independently distributed as $\mathcal{N}(0, \sigma^2)$. We thus have that

$$Z_{\beta'}(B) = \sum_{i=1}^n \left((\beta' - \beta^*)^\top X^{(i)}\right)^2 - \left((\beta - \beta^*)^\top X^{(i)}\right)^2 + 2\xi^{(i)} \ (\beta - \beta')^\top X^{(i)}.$$

Denote the three terms in the summation by:

$$Z_1 := \sum_{i=1}^n \left((\beta' - \beta^*)^\top X^{(i)}\right)^2, Z_2 := \sum_{i=1}^n \left((\beta - \beta^*)^\top X^{(i)}\right)^2, Z_3 := \sum_{i=1}^n 2\xi^{(i)} \ (\beta - \beta')^\top X^{(i)}.$$

We proceed to argue that $Z_1$ is bounded from below, and $Z_2, Z_3$ are bounded from above with high probability. [5].

For $Z_1$, applying the weak anti-concentraiton property of $X$ (Corollary E.5) gives that

$$\Pr\left[\left((\beta' - \beta^*)^\top X^{(i)}\right)^2 \geq \|\beta' - \beta^*\|_2^2/2\right] \geq \gamma.$$

for some universal constant $\gamma$. By the Chernoff bound, given that $n \gg \log(1/\alpha)$, the fraction of $X^{(i)}$ such that $\left((\beta' - \beta^*)^\top X^{(i)}\right)^2 \geq \|\beta' - \beta^*\|_2^2/2$ will be at least $\gamma/2$ with probability at least $1 - \alpha/120$. It then follows that

$$\Pr\left[Z_1 \leq \frac{\gamma n}{4}\|\beta' - \beta^*\|_2^2\right] \geq 1 - \alpha/120. \tag{19}$$

For $Z_2$, since $\mathbf{E}[X^{(i)} \left(X^{(i)}\right)^\dagger] = I$ by Assumption 1.2, it follows that $\mathbf{E}[Z_2] = n\|\beta - \beta^*\|_2^2$. In order to show that $Z_2$ is sufficiently concentrated, we will bound from above the $k$-th central moments of $Z_2$ for some even integer $k \leq \Delta$. Define

---

[5]Note that there are correlations between $Z_1, Z_2, Z_3$. Nonetheless, these correlations will not affect our analysis.

$y_i = \left( (\beta - \beta^*)^\top X^{(i)} \right)^2$. We note that the $y_i$s are i.i.d. random variables with their degree-$k$ central moments bounded from above by

$$\mathbf{E}\left[ \left( \left( (\beta - \beta^*)^\top X^{(i)} \right)^2 - \|\beta - \beta^*\|_2^2 \right)^k \right] \leq 2^k \mathbf{E}\left[ \left( (\beta - \beta^*)^\top X^{(i)} \right)^{2k} + \|\beta - \beta^*\|_2^{2k} \right]$$

$$\leq 2^{k+1} \|\beta - \beta^*\|_2^{2k} Q$$

where in the first line we apply the triangle inequality (Fact B.2), and in the second line we use the assumption on the degree $2k$ moments of $X$. Hence, applying Lemma 3.3 gives that the degree $k$ moment of $Z_2$ is bounded from above by

$$\mathbf{E}\left[ \left( \sum_{i=1}^{n} (y_i - \mathbf{E}[y_i]) \right)^k \right] \leq 2 \, (4kn)^{k/2} \, \|\beta - \beta^*\|_2^{2k} \, Q \; .$$

In other words, we have that

$$\left( \mathbf{E}\left[ \left( \sum_{i=1}^{n} (y_i - \mathbf{E}[y_i]) \right)^k \right] \right)^{1/k} \leq 2^{1/k} \sqrt{4kn} \, Q \|\beta - \beta^*\|_2^2.$$

By Chebyshev's inequality, we thus have that

$$\Pr\left[ Z_2 \geq \|\beta - \beta^*\|_2^2 \left( n + 100\sqrt{kn} \, Q^{1/k} \right) \right] \leq \alpha/120. \tag{20}$$

For $Z_3$, note that $\mathbf{E}[Z_3] = 0$ since $\xi^{(i)}$ has mean 0. To argue for its concentration, we again proceed to bound its degree-$k$ moment for some even integer $k$. Similarly, we define $z_i = \xi^{(i)} (\beta - \beta')^\top X^{(i)}$. The degree-$k$ central moments of $z_i$ can be bounded from above by

$$\mathbf{E}\left[ \left( \xi^{(i)} (\beta - \beta')^\top X^{(i)} \right)^k \right] = \mathbf{E}\left[ \left( \xi^{(i)} \right)^k \right] \mathbf{E}\left[ \left( (\beta - \beta')^\top X^{(i)} \right)^k \right].$$

We can apply the upper bounds on the degree-$k$ moments of $\xi^{(i)}$ and $X^{(i)}$ respectively. This allows us to conclude that

$$\mathbf{E}\left[ \left( \xi^{(i)} (\beta - \beta')^\top X^{(i)} \right)^k \right] \leq k^{k/2} Q \sigma^k \|\beta - \beta'\|_2^k.$$

Applying Lemma 3.3 then gives that

$$\mathbf{E}\left[ \left( \sum_{i=1}^{n} z_i \right)^k \right] \leq n^{k/2} \, k^k Q \sigma^k \|\beta - \beta'\|_2^k.$$

In other words, we have that

$$\left( \mathbf{E}\left[ \left( \sum_{i=1}^{n} z_i \right)^k \right] \right)^{1/k} \leq k\sqrt{n} \, \sigma \, \|\beta - \beta'\|_2 Q.$$

By Chebyshev's inequality, it holds that

$$\Pr\left[ Z_2 > 10k\sqrt{n} \, \sigma Q \, \|\beta - \beta'\|_2 \right] \leq \alpha/120. \tag{21}$$

By the union bound, the events in Equation (19), Equation (20), and Equation (21) are satisfied simultaneously with probability at least $1 - \alpha/40$. When that happens, $Z_{\beta'}(B)$ will be bounded from below by

$$\frac{\gamma}{4} \, n \, \|\beta' - \beta^*\|_2^2 - \|\beta - \beta^*\|_2^2 \, \alpha^{-1/k} \left( n + 100\sqrt{kn} \, Q^{1/k} \right) - 10k\sqrt{n} \, \sigma Q \, \alpha^{-1/k} \, \|\beta - \beta'\|_2. \tag{22}$$

First, we claim that

$$\|\beta' - \beta^*\|_2 \gg \|\beta - \beta^*\|_2 \tag{23}$$
$$\|\beta' - \beta^*\|_2 \geq (1 - o(1))\|\beta - \beta'\|_2. \tag{24}$$

To prove Equation (24), we note that

$$\|\beta' - \beta^*\|_2 \geq \|\beta' - \beta\|_2 - \|\beta - \beta^*\|_2 \geq (1 - o(1))\|\beta - \beta'\|_2.$$

where the first inequality is the triangle inequality, and the second inequality is true by our assumption that $\|\beta - \beta^*\|_2 < R \ll \|\beta' - \beta\|_2$. Equation (23) then follows immediately as $\|\beta - \beta'\|_2 \gg \|\beta - \beta^*\|_2$.

With the above inequalities in mind, we proceed to argue that the positive term dominates all the negative terms in Equation (22). Since $\gamma$ is a universal constant, it follows that

$$\gamma n \, \|\beta' - \beta^*\|_2^2 \gg \|\beta - \beta^*\|_2^2 n.$$

Next recall that $n \gg k \, Q^{2/k} \, \alpha^{-2/k}$ by our assumption on $n$. It then follows that

$$\gamma n \, \|\beta' - \beta^*\|_2^2 \gg \|\beta - \beta^*\|_2^2 \, 100\sqrt{kn} \, Q^{1/k} \, \alpha^{-1/k}.$$

Lastly, recall that we assume $\|\beta - \beta'\|_2 \gg k\sigma Q \, \alpha^{-1/k}/\sqrt{n}$. Combining this with Equation (24) and Equation (23) then gives that $\|\beta' - \beta^*\|_2 \geq (1 - o(1)) \, \|\beta - \beta'\|_2 \gg k\sigma Q \, \alpha^{-1/k}/\sqrt{n}$, which implies that $\|\beta' - \beta^*\|_2^2 \gg \|\beta - \beta'\|_2 k\sigma Q \, \alpha^{-1/k}/\sqrt{n}$. It then follows that

$$\gamma n \, \|\beta' - \beta^*\|_2^2 \gg 10k\sqrt{n} \, \sigma Q \, \alpha^{-1/k} \, \|\beta - \beta'\|_2$$

Combining the above gives that

$$\Pr_{B \sim \mathcal{D}_{\beta^*}} [Z_{\beta'}(B) > 0] > 1 - \alpha/40 \, , \tag{25}$$

as long as $n \gg k \, Q^{2/k} \, \alpha^{-2/k}$ and $\|\beta - \beta'\|_2 \gg R + k\alpha^{-1/k}\sigma Q^{1/k}/\sqrt{n}$.

Since the inlier batches are all drawn independently, it holds the faction of inlier batches violating the condition is at most $\alpha/20$ with probability at least $1 - \delta/K^2$ when the number of inlier batches drawn are at least $N \gg \log(K/\delta)\alpha^{-2}$. Since the size of $L$ is at most $K$, there are at most $K - 1$ many $\beta'$ we need to consider. Condition 6 is therefore satisfied with probability at least $1 - \delta$ by the union bound.

When we have $\log(K) > d^2$, we will need an alternative argument. We note that $Z_1$ and $Z_2$ are both linear functions in the random variables $\sum_{i=1}^{n} X^{(i)} X^{(i)^T}$ of dimension $d^2$, and $Z_3$ is a linear function in the random variables $\sum_{i=1}^{n} \xi^{(i)} X^{(i)}$. Thus, overall, for any $\beta' \in \mathbb{R}^d$, $Z_{\beta'}(B)$ is a linear function in $O(d^2)$ many random variables. It then follows that, for any $\beta' \in \mathbb{R}^d$, $\mathbb{1}\{Z_{\beta'}(B) > 0\}$ is an $O(d^2)$-dimensional linear threshold function, which has VC-dimension $O(d^2)$. Let $G$ be $N' \gg d^2\alpha^{-2}\log(1/\delta)$ many inlier batches drawn from $D_{\beta^*}$. By the VC-inequality, we thus have

$$\Pr_G \left[ \sup_{\beta' \in \mathbb{R}^d} \left| \Pr_{B \sim G} [Z_B(\beta') > 0] - \Pr_{B \sim D_{\beta^*}} [Z_B(\beta') > 0] \right| > \alpha/20 \right] \leq \delta.$$

Combining this with Equation (25) then shows that Condition 6 is satisfied with probability at least $1 - \delta$. $\qquad\square$

# F. Reduction from the Batch-Setting to the Non-Batch Setting

We point out a simple reduction (in Claim F.1), which allows one to solve list-decodable linear regression in the non-batch setting using an algorithm for the batch-setting in a black-box manner. The idea is the trivial observation we can construct our own batches of size $n$ just by collecting together $n$ individual labeled examples. Denote by $\alpha$ the probability that an individual example is inlier. Then the probability that a batch made in the aforementioned way consists only of inliers is $\alpha_B = \alpha^n$. Then, running any algorithm designed for the batch setting should yield guarantees where the corruption rate is being replaced by $\alpha^n$. In particular, if we denote by $m(\alpha_B, d), \ell(\alpha_B)$ and $\text{error}(\alpha_B)$ the sample complexity, list size and error guarantee of the black-box algorithm (which are functions of the corruption rate $\alpha_B$ and maybe other parameters

like the dimension $d$ which do not matter for this discussion), then the resulting algorithm for solving the problem in the non-batch setting will have its sample complexity, lits size and error rate being $m(\alpha^n, d)$, $\ell(\alpha^n)$ and $\text{error}(\alpha^n)$ respectively.

For convenience, throughout this section we will restrict our ourselves to the case $\alpha < 1/2$, which corresponds to more than half of the data being corrupted. We are interested only in this since this is the truly "list-decodable setting". For this reason, we will use $n \ll \log(1/\alpha_B)$ in the claim below (because we have already mentioned that $\alpha = \alpha_B^{1/n}$, thus in order to have $\alpha < 1/2$ we need $n \ll \log(1/\alpha_B)$).

**Claim F.1.** *Denote by $d$ the ambient dimension and by $\alpha \in (0, 1/2)$ the corruption level for the non-batch setting. Let $c > 0$ be a sufficiently small absolute constant. Suppose that $\mathcal{A}$ is an algorithm with the guarantee that for any $\alpha_B \in (0, 1/2)$ it can draw $m(\alpha_B, d)$ batches of size $n = c\log(1/\alpha_B)$ from the corrupted distribution of Definition 1.1 with corruption level $\alpha_B$, and output a list of size $\ell(\alpha_B)$ of vectors which contains a vector $\hat{\beta}$ with $\|\hat{\beta} - \beta^*\|_2 \leq \text{error}(\alpha_B)$. Then, there exists another algorithm $\mathcal{A}'$ that draws $m(\alpha^n, d)$ batches of size $1$ from the corrupted distribution of Definition 1.1 with rate of corruption $\alpha$, and outputs a list of size $\ell(\alpha^n)$ of vectors which contains a vector $\hat{\beta}$ with $\|\hat{\beta} - \beta\|_2 \leq \text{error}(\alpha^n)$.*

This reduction, in combination with the lower bound of (Diakonikolas et al., 2021), can serve as informal evidence that doing list-decodable linear regression with batch sizes $n \ll \log(1/\alpha_B)$ likely requires exponential time. In particular, Theorem 1.5 in (Diakonikolas et al., 2021) provides evidence[6] that any algorithm with polynomial sample complexity needs exponential list-size or exponential runtime. Let $n = c\log(1/\alpha_B)$ for some constant $c \ll 1$. If Theorem 1.3 were to allow for that batch size of $n = c\log(1/\alpha_B)$ (recall that it right now only works for $n \gg \log(1/\alpha_B)$), then, by the reduction above (Claim F.1) we would obtain an algorithm for the non-batch setting, with quasi-polynomial runtime and list size which would contradict the hardness evidence.

---

[6]By "evidence" we mean that the lower bound is only about the Statistical Query model. Although this does not imply hardness results for the class of all algorithms, SQ lower bounds have long served as strong indication of hardness.

