# OpenReview forum: "Batch List-Decodable Linear Regression via Higher Moments"
_ICML.cc/2025/Conference — ICML 2025 poster_

### Official Review · Reviewer_gZA8 · 2025-03-07

**Overall Recommendation:** 4

**Summary:**

In modern machine learning, collecting large datasets from a single source is often impractical. Instead, data is typically gathered in batches from multiple sources. This paper investigates the $\textbf{batch list-decodable linear regression}$ problem: given pairs $(X, y) \in \mathbb{R}^{d+1}$ drawn from the distribution $D_{\beta^*}$ such that $y = {\beta^*}^\top X + \xi$, where $\xi \sim \mathcal{N}(0, \sigma^2)$, we are provided with $m$ batches, each containing $n$ samples. With probability $\alpha$, a batch consists entirely of i.i.d. samples from $D_{\beta^*}$, while with probability $1-\alpha$, the batch is drawn from an arbitrary distribution. The goal is to compute a list $L$ of vectors $\widehat{\beta}$ such that at least one $\widehat{\beta}$ satisfies that $\\|\widehat{\beta} - \beta^*\\|_2$ is small.
Compared to the prior work [DJKS'23], which reduced the batch size dependency from exponential in $1/\alpha$ to linear, this paper further improves the batch size and achieves lower estimation error by leveraging the higher-order moment assumptions through the Sum-of-Squares (SoS) framework.

**Claims And Evidence:**

Yes, the claims in this paper are clearly stated and rigorously supported by detailed proofs.

**Essential References Not Discussed:**

None.

**Experimental Designs Or Analyses:**

No experiments in this paper.

**Methods And Evaluation Criteria:**

The main result of this paper is purely theoretical and is thoroughly validated through rigorous theoretical analysis.

**Other Comments Or Suggestions:**

It would clarify the generality of the proposed method if this paper could discuss the tightness of the SoS assumptions.

**Other Strengths And Weaknesses:**

$\textbf{Strengths:}$ (1) The proposed algorithm improves the performance of previous works by requiring smaller batch sizes and achieving better error guarantees, and provides a flexible tradeoff between batch size and computational complexity.
(2) This paper leverages higher-order moment information under the SoS framework to improve the robustness and accuracy of the estimation.
(3) This paper provides the first SoS proof of the Marcinkiewicz-Zygmund Inequality, which could be useful in other robust estimation tasks.
$\textbf{Weaknesses:}$ (1) The proposed method relies on SoS-certifiably bounded moments, which might limit its applicability to more general distributions.
(2) The algorithm is polynomial in the dimension and batch size but can become quasi-polynomial for very small batch sizes.

**Questions For Authors:**

This paper is closely related to the work of Das et al. [DKS'23]. Could you compare the running time of the proposed algorithm with that of [DKS'23]?

**Relation To Broader Scientific Literature:**

(1) The use of higher moments and SoS-based techniques could be applied to other robust regression settings, such as sparse linear regression or generalized linear models, where outliers or heavy-tailed noise are common.
(2) The batch setting considered in this paper shares similarities with mixed linear regression, where each batch could correspond to a different component of the mixture. Extending the SoS-based approach to estimate multiple regression components simultaneously would be an interesting direction.
(3) The SoS framework might be leveraged to design robust algorithms for list-decodable PCA under the batch setting, especially in the presence of corruptions or outliers.

**Theoretical Claims:**

The proofs seem correct since I didn't read all the proofs carefully.

---

> ### Author Rebuttal · Authors · 2025-04-01
>
> We thank the reviewer for their effort and positive assessment of our work. We reply to the points mentioned individually below:
>
> (**Applicability of the SoS-certifiability Assumption**) SoS certifiability of moments is by now a well-studied condition that is known to hold for broad families of interest such as sub-Gaussian distributions and all distributions satisfying the Poincare inequality. Please see our response to Reviewer ez6k on this.
>
> (**Comparison of runtime with Das et al. 2023**) As discussed in lines 91-107, the runtime of Das et al. 2023 is $\mathrm{poly}(d,n,m,1/\alpha)$. Their theorem statement does not specify what is the absolute constant in the exponent of that runtime. In comparison, our algorithm is also polynomial time but the parameter $k$ appears in the exponent (order of the higher moment information used) instead of just an absolute constant.
>
> (**Quasi-polynomial runtime for very small batch sizes**) It is indeed true that the algorithm is no longer polynomial-time when the batch size becomes very small (e.g., when $k$ is on the order of $\log^2(1/\alpha)$). As discussed after the main theorem statement (lines 115–131), there is evidence (based on previously known Statistical Query lower bounds from Diakonikolas et al. 2021) that this phenomenon is inherent. We view this as an interesting conceptual conclusion: The tradeoff between complexity and batch size appears to be smooth. Specifically, for $n = 1/\alpha$, the complexity of our algorithm matches the previously known algorithm of Das et al., 2023, while for $n = \log^2(1/\alpha)$, it matches known SQ hardness results. In the intermediate regime, our results suggest a continuous interpolation rather than a sharp phase transition at a specific point.

---

### Official Review · Reviewer_GB1B · 2025-03-08

**Overall Recommendation:** 4

**Summary:**

First, in the interest of full disclosure, this review is very lightly modified from a review I wrote for this paper for NeurIPS 2024.

This paper studies a problem in algorithmic robust statistics: batch list-decodable linear regression. The setting is as follows. We get $m$ batches of $n$ samples $(X_i,Y_i)$ each, from a linear model $Y_i = X_i^\top \beta + \epsilon_i$. An $\alpha > 0$-fraction of the batches are "good", ie distributed like above. The remaining $(1-\alpha)m$ batches are actually chosen by a malicious adversary. The goal is to output a list of possible $\beta$s, actually $O(1/\alpha) of them (which is the best you can hope for), one of which is close to the ground truth.

The motivation for this problem comes from:

- list-decodable regression *without* the batch assumption -- ie just $1/\alpha$ fraction of samples are "good" -- seems to require exponential in $1/\alpha$ time, per SQ lower bounds
- in the real world, datasets are often collected in batches, but the batches may be small.

The paper's main contribution is a new family of polynomial-time algorithms with stronger quantitative guarantees than prior work for this problem. The key improvement is to the size of batches needed. Prior works obtained provable guarantees in polynomial time only when the batch size $n \gg 1/\alpha$. By contrast, the present work tolerates really small batches, of arbitrarily small polynomial size -- $n \gg \alpha^{-\delta}$ for any small constant $\delta > 0$. The tradeoff is that the resulting algorithm requires:

- strong assumptions on $X$: the random variable $X$ for the "good" samples must be "SoS-certifiably bounded" for moments up to $O(1/\delta)$. This assumption is satisfied by strongly log-concave distributions and by subgaussian distributions; it is largely unknown what other distributions may satisfy it. (The paper should probably mention the fact that subgaussian distributions satisfy their assumption.)
- a lot of samples and time: (nd)^{O(1/\delta)}$ batches and time are needed.

The algorithm and its analysis borrow a lot of technical ingredients from the now-enormous algorithmic robust statistics literature. It is a little unclear to me to what extend there is a fundamental new algorithmic/analysis idea here, although certainly combining several existing techniques from algorithmic robust stats is on its own a nontrivial matter. The authors suggest that their combination of an iterative list-pruning technique with SoS is novel; I am not sufficiently expert in the specifics of list-decodable estimation to assess the novelty independently.

**Claims And Evidence:**

yes

**Essential References Not Discussed:**

none

**Experimental Designs Or Analyses:**

n/a

**Methods And Evaluation Criteria:**

yes

**Other Comments Or Suggestions:**

none

**Other Strengths And Weaknesses:**

## Strengths

Pushes the frontier of polynomial-time guarantees for a well-motivated problem in robust statistics. Reasonably well written.

## Weaknesses

While batch list-decodable linear regression itself seems like a reasonably well-motivated problem, I see only moderate motivation for the goal of obtaining algorithms tolerating smaller batch size if the resulting algorithms have impractical (albeit still polynomial) running time. Given that algorithmic robust statistics has been under intense investigation for almost a decade now, I think a strong new paper should ideally either:

1. introduce a fundamental new algorithmic or lower-bound technique, overcome a serious technical barrier, etc., and/or
2. make headway in pushing the many new theoretical ideas from the last 10 years towards the realm of practical algorithms/impact beyond theory.

I think this paper is making some contribution on both types of goals, but not groundbreaking contribution on either. For (1), the paper is indeed pushing the frontier of polynomial-time algorithms, but it is hard to tell if there's a really new technique here, versus clever combination of existing ideas. For (2), although pushing for smaller batch sizes is a well-motivated practical goal, the paper is hamstrung by its reliance on higher-moment techniques and SoS which give very large polynomial running time.

Still, I think the problem is appealing enough and the results will be appreciated by the ICML audience.

**Questions For Authors:**

none

**Relation To Broader Scientific Literature:**

thoroughly discussed in the introduction to the paper; I will not repeat the discussion here.

**Theoretical Claims:**

no

---

> ### Author Rebuttal · Authors · 2025-04-01
>
> We thank the reviewer for their time and the positive evaluation of our paper. We respond to the points raised individually below:
>
> (**Assumptions on $X$**) The SoS certifiability of moments is a well-established condition, known to hold for various important distribution families, including sub-Gaussian distributions and those satisfying the Poincaré inequality. Please see our response to Reviewer ez6k for more details.
>
> (**Sample and Time Complexity**) There is a tradeoff between the batch size and the complexity of the algorithm: Before this work, we only knew that for batch size $n=1$ the problem exhibits exponential complexity hardness results and for $n=1/\alpha$ it becomes feasible in polynomial time, but we did not know anything in between: Is the complexity interpolating in a smooth way or there is some phase transition phenomenon? One of the main goals of this paper is to quantify this tradeoff and show that it is indeed smooth, which we view as an interesting theoretical insight. As explained in lines 114-131 (first column) although this tradeoff means that for extremely small delta the algorithm becomes efficient, this matches the prior known SQ lower bounds, thus in this sense, this behavior is unavoidable. Moreover, there is a wide range of regimes where the algorithm is efficient. For example setting $\delta$ to be a constant like $¼$, already gives a polynomial time algorithm that works only under 4-th bounded moment distributions and has better error guarantee than prior work (any algorithm before our work would either have larger error or run in exponential time).
>
> (**Technical contributions**) One of our main technical contributions is a novel pruning procedure to reduce the candidate list size to $O(1/\alpha)$. This procedure is optimal, and uses techniques that differ significantly from prior work. The procedure is critical for the success of the iterative algorithm as the number of candidate lists will grow exponentially with the number of iterations without pruning. Moreover, although robust mean estimation using SoS is by now a standard tool, our work requires a novel iterative framework where each iteration improves the accuracy of the estimators. Lastly, in order to apply the result from Kothari & Steinhardt (2017), we need to show that the fact that the covariate has certifiably bounded moments implies that the regressor estimator we use also has SoS certifiably bounded moments, which is technically non-trivial. Finally, in order to apply SoS robust mean estimators for the variable $yX$ we need to ensure certifiability of moments, for the purpose of which we prove an SoS version of the famous Marcinkiewicz Zygmund Inequality, that might be useful independently of the result of this paper.
>
> (**Motivation of Studying Batch-size in Robust Statistics Tasks**) Once again, we would like to emphasize the importance of studying the role of batch size in robust statistics. While the common assumption in robust statistics is that an arbitrary subset of data points (with a certain size bound) is corrupted, in the crowdsourcing setting, adversarial corruption naturally happens at a user-level, leading to corrupted batches of data instead of corrupted data points. As shown in Diakonikolas et al. (2021) and Das et al. (2023), such batch structure in corruption may lead to drastic change in the computation landscape of the statistical estimation task, and we believe it is a natural as well as important research direction to explore the quantitative relationship between batch size and other algorithmic resources needed in robust estimation tasks.  Our work is the first revealing a smooth quantitative tradeoff between batch sizes and the computation/statistical resources needed for the concrete application of list-decodable linear regression.
>
> On the practical side, we believe that optimizing the batch-size is crucial to any crowd-sourcing setting. In particular, if only 1% of users were reliable, prior work would roughly require one of the following: Either each user provides one data point, but the learning algorithm has exponential complexity Karmalkar et al. (2019), or (ii) each user will have to contribute at least $100$ data points Das et al. (2023) (which may well be unrealistic). On the other hand, we give an efficient algorithm that succeeds with much fewer data points per user; prior to our work, it was not clear if this is even possible in polynomial time. In particular, even if we use only the first $k=4$-th order of moment information, our algorithm will have already improved upon the prior work with substantially smaller batch sizes and achieved better error. We thus view our work as an important first step towards building practical algorithms for smaller batch sizes.

---

### Official Review · Reviewer_ez6k · 2025-03-12

**Overall Recommendation:** 4

**Summary:**

This paper proposes an efficient polynomial-time algorithm for batch list-decodable linear regression, using higher-order moment information within a Sum-of-Squares (SoS) framework. Compared to previous methods, this approach notably reduces both the required minimum batch size and the final regression error. These improvements rely on assuming that the covariates have higher-order moments that are certifiably bounded within the SoS framework. The authors' main innovation is an iterative list-pruning algorithm that leverages these higher moments, enabling stronger performance guarantees than existing techniques.

**Claims And Evidence:**

The claims are generally well-supported by rigorous theoretical arguments.

**Essential References Not Discussed:**

This paper studies batch list-decodable linear regression via higher moments, where the list-decodable linear regression is provided by [2].

[2] Sushrut Karmalkar, Adam R. Klivans, Pravesh Kothari: List-decodable Linear Regression. NeurIPS 2019: 7423-7432

**Experimental Designs Or Analyses:**

This paper focuses exclusively on theoretical contributions without empirical validation.

**Methods And Evaluation Criteria:**

The proposed methods and/or evaluation criterias are sound and particularly suitable.

**Other Comments Or Suggestions:**

It will be better if the authors add numerical experiments to validate your theories.

**Other Strengths And Weaknesses:**

Strengths:
1. Strong theoretical contribution: notably improves upon previous algorithms by leveraging higher moments.
2. Introduction of novel theoretical techniques (SoS Marcinkiewicz-Zygmund inequality) valuable to the broader theoretical ML community.

Weaknesses:
1. The assumptions required for significant improvements (e.g., hypercontractivity, bounded SoS-certifiable moments) might limit broader applicability.
2. No experimental validation, even preliminary, to provide intuition about practical relevance.

**Questions For Authors:**

See weaknesses.

**Relation To Broader Scientific Literature:**

The paper places itself clearly within the literature, comparing explicitly and thoroughly with prior works (Das et al. 2023, Diakonikolas et al., 2019, Diakonikolas & Kane, 2023, Kothari & Steinhardt, 2017). It effectively leverages and builds upon known techniques from robust and list-decodable regression, robust mean estimation, and the SoS methodology. It makes a clear advancement in terms of algorithmic complexity relative to Das et al. (2023).

**Theoretical Claims:**

The key theoretical contributions—particularly Lemma 3.3 (SoS Marcinkiewicz–Zygmund inequality)—are clearly and rigorously presented. Additionally, I checked Theorem 1.3 and Proposition 3.1, whose proof techniques are based on reference [1]. The provided proofs appear correct, though very technical.

[1]. Das, A., Jain, A., Kong, W., and Sen, R. Efficient list decodable regression using batches. In International Conference on Machine Learning, pp. 7025–7065. PMLR, 2023.

---

> ### Author Rebuttal · Authors · 2025-04-01
>
> We thank the reviewer for their time and positive assessment of our work. We respond to their points below:
>
> (**Distributional assumptions**) The condition of having SoS certifiably bounded moments is standard (in the algorithmic robust statistics literature) for leveraging higher-order moment information in the algorithm. This condition holds for all distributions satisfying the Poincare inequality (Kothari & Steinhardt (2017)), which results in a large family of distributions: it includes Gaussians, product distributions, strongly log-concave distributions, and any sum or uniformly continuous transformation of such distributions. There are also other non-Poincare distributions with SoS certifiably bounded moments, i.e., distributions over discrete points sampled from Gaussians. Moreover, via recent results (Diakonikolas et al. (2024) cited at the end of this rebuttal), it is known that the class of all sub-Gaussian distributions also satisfies the SoS certifiably bounded moments condition. Finally, regarding practicality, if having bounded $k$-th moments for very large $k$ is viewed as unrealistic in practice, we point out that even if we use only the first $k=4$-th order of moment information, our algorithm will have already improved upon the prior work with substantially smaller batch sizes and achieved better error. We thus view our work as an important step towards building practical algorithms for smaller batch sizes.
>
> (**Experimental evaluation**) While we acknowledge the reviewer's interest in experiments and the importance of developing practical algorithms, we would like to emphasize that our primary contribution is to characterize the computational-statistical landscape for this fundamental learning task—in terms of error guarantee, batch size complexity, and batch number, and the tradeoff between them. That being said, there are practical regimes (like $k=4$) where we already get improved error compared to Das et. al. 2023 in poly-time and any other previously known alternative for the same error would require exponential runtime.
>
> References:
>
> Diakonikolas, Ilias, Samuel B. Hopkins, Ankit Pensia, and Stefan Tiegel. "SoS certifiability of subgaussian distributions and its algorithmic applications." arXiv preprint arXiv:2410.21194 (2024).

---

### Official Review · Reviewer_HmAU · 2025-03-14

**Overall Recommendation:** 3

**Summary:**

This paper studies the list-decodable linear regression, under the setting that the algorithm can collect batches of samples. This paper can be seen as a follow-up of Das et al. 2023, which studies the same problem under same batch setting, however uses a batch size $n \geq \tilde{\Omega}(\alpha^{-1})$, number of batches $m=\text{poly}(d,n,\alpha^{-1})$, and outputs a list of $O(\alpha^{-2})$ vectors at least one of which is $\tilde{O}(\alpha^{-1/2}/\sqrt{n})$ close to the target regressor. This paper, proposing a new algorithm that uses degree-$1/\delta$ moment information, enjoys a batch size $n\geq\Omega_{\delta}(\alpha^{-\delta})$ with $m = \text{poly}((dn)^{1/\delta},\alpha^{-1})$ and outputs a list of $O(\alpha^{-1})$ vectors at least one of which is $O(\alpha^{-\delta/2}/\sqrt{n})$ close to the target. The algorithm implements a refining idea for regressor using mean estimation from Diakonikolas et al. (2019) and apply a SoS list-decodable mean estimation approach to handle the linear regression in list-decodable setting.

**Claims And Evidence:**

Given that the prior work on batch list-decodable linear regression didn't use high-moment information, the improvement makes senses in some way.

However, the algorithm is somehow not clear, maybe due to the complexity of this problem and SoS based algorithms. As stated in the paper, the regressor is estimated using the mean of $yX$, and the list-decodable learning is made possible via implementation of SoS based algorithms for mean estimation. Then, it should be clearly specify which mean estimation framework the algorithm is based on, as the literature for list-decodable mean estimation is well developed. On the other hand, it is not true that there exists no multi-filter framework using high-moment information. Actually, the very beginning of the multi-filter framework proposed by Diakonikolas et al. 2018 already considered higher moments. Hence, I don't see what the challenging part is here.

**Essential References Not Discussed:**

It seems all are cited.

**Ethical Review Concerns:**

No concerns.

**Experimental Designs Or Analyses:**

No experiments required.

**Methods And Evaluation Criteria:**

The method makes sense, by applying robust linear regression technique together with a list-decodable mean estimator.

**Other Comments Or Suggestions:**

It must be specify the tradeoff between batch size and number of batches.

**Other Strengths And Weaknesses:**

There is a key weakness for this paper. As the key contribution is the improvement over the batch size, however, this is at the cost of increasing the number of batches, which is not stated in this paper. The paper nor discusses the total sample complexity and how it is affected by applying different parameter $k$ in its algorithm. To check this closely, although batch size is reduced to $n=\Omega_{\delta}(\alpha^{-\delta})$, the number of batches $m=\text{poly}((nd)^{1/\delta},\alpha^{-1})$. Hence, to avoid higher number of batches, if $\delta=1$, the result is essentially the same to Das et al. 2023. Since there are lots of hiding terms, it is hard to verify if this algorithm is truly efficient.

**Questions For Authors:**

See above.

**Relation To Broader Scientific Literature:**

This paper aims to improve upon previous work on batch list-decodable linear regression. The findings on establishing a SoS proof for the Marcinkiewicz Zygmund Inequality, might be of broader interest.

**Theoretical Claims:**

The theoretical claims make sense. It is hard for me to check all of the proofs.

---

> ### Author Rebuttal · Authors · 2025-04-01
>
> We thank the reviewer for their time and effort in reading our paper. We respond to the points raised individually below:
>
> (**Tradeoff between batch size and algorithm complexity is not discussed**) There indeed is a tradeoff between batch size and algorithm complexity. However, this is not a hidden aspect of the result. Rather, it is explicitly stated, and it is part of the conceptual take-away point of the paper. We reiterate the main points below for completeness:
> The tradeoff is clearly presented in Theorem 1.3, where all parameters (batch size, number of batches, and runtime) are formally stated. Specifically, the batch size $n = \Omega_{\delta}(\alpha^{-\delta})$ and the number of batches $m = \text{poly}((nd)^{1/\delta}, \alpha^{-1})$, referenced by the reviewer, are explicitly given there.
> The tradeoff is discussed in detail directly after the theorem (lines 93-131). Specifically the discussion paragraph states “Our result essentially shows that there is a smooth tradeoff between the batch size provided and the computational resources required”. The subsequent text points out that if the parameter $\delta$ is superpolynomial then the algorithm is no longer efficient and that there is evidence in the form of known Statistical Query lower bounds, (discussed further in Appendix F) suggesting that this tradeoff is unavoidable.
> Quantifying this tradeoff is part of the main conceptual contribution of the work: On the one hand, SQ lower bounds from Diakonikolas et al., 2021 suggest that the problem is hard for batch-size equal to one; while Das et al. (2023) gives an efficient algorithm when the batch-size is $\approx 1/\alpha$ (see lines 69-81). It is natural to ask what happens in between (lines 91-97). Is there a sharp phase transition (in terms of the existence of a poly-time algorithm) when the batch-size goes from constant to linear in $1/\alpha$ or does the tradeoff happen smoothly? Our result suggests the latter: when the batch-size is $1/\alpha^c$, for $c>0$, there exists an algorithm using $O(1/c)$-degree SoS to achieve a small error for the task (lines 112-131).
> If the reviewer’s concern stems from our use of the term “efficient”, such as in lines 104–108 (“Is there a computationally efficient algorithm […] with improved batch size and/or error guarantees?”), we clarify that “efficient” there refers to poly-time algorithms in general, which is the standard convention in the learning theory literature. This does not imply the absence of trade-offs within the space of poly-time algorithms. If this was the source of confusion, we will revise the phrasing to ensure this distinction is explicit.
>
> (**Clarifications on the algorithmic approach**) We respond to the specific questions below:
> * (*Which mean estimation framework is being used?*) As stated in line 166 of our paper, either Theorem 5.5 from Kothari & Steinhardt (2017), or Theorem 6.17 from Diakonikolas & Kane (2023) suffices for our purpose of list-decodable mean estimation. In general, any list-decodable mean estimation algorithm satisfying the guarantee in Lemma 3.5 suffices for our purpose, and the exact implementation is not important for our algorithm.
> * (*“It is not true that there exists no multi-filter framework using high-moment information. Actually, the very beginning of the multi-filter framework proposed by Diakonikolas et al. 2018 already considered higher moments”*). The proof strategy in Diakonikolas et al. 2018 requires exact Gaussianity of the data (also listed in the their theorem statement). The subsequent work of Diakonikolas et al. 2020 was able to adapt the proof strategy for the case of just bounded covariance data but it did not extend for higher moment boundedness. We refer to Chapter 6 of the book Diakonikolas & Kane (2023) where the difficulties of using higher information is described. To the best of our knowledge SoS based algorithms are the only ones that work with higher moments beyond Gaussian distributions. Thus, we apply an SoS list-decodable mean estimation to the regressor estimator $\frac{1}{|B|}\sum_{(X,y) \sim B} Xy$. Even if $X$ were Gaussian, this estimator would not be, making Diakonikolas et al. (2018) inapplicable. To address this, we show SoS certifiability of the moment bounds by deriving a novel SoS proof of the Marcinkiewicz-Zygmund inequality that may be of independent interest. Finally, we  highlight that the biggest difficulty in our paper is not finding the right algorithm to apply as a black-box but the subsequent issue that applying this algorithm iteratively may result in a number of hypotheses that blows up (see the paragraphs after line 194). Making that iterative framework work is where our novel pruning procedure is used. This procedure is optimal, and uses techniques that differ significantly from prior work of Diakonikolas et al. 2020.
>
> I. Diakonikolas,  D. M. Kane, and A. Stewart. List-decodable robust mean estimation and learning mixtures of spherical gaussians. STOC 2018.

---

> > ### Comment · Reviewer_HmAU · 2025-04-07
> >
> > Thanks for the response!
> >
> > The tradeoff:
> >
> > Indeed, my concern is that when $\delta$ is no longer a constant, the the claim of ``computationally-efficient'' does not hold any more, as there is a $1/\delta$ factor in the exponent of the number of batches. I think this must be clarified when claiming computational efficiency. As the main question addressed in Line 105-107 is the existence of computationally-efficient algorithm with 'significantly improved' batch size. It is unclear whether this `significant improvement' is qualitative or quantitative. To claim efficiency, $\delta$ must be an absolute constant, meaning that the batch size is still $\text{poly}(1/\alpha)$.
> >
> > On the other hand, I do appreciate the contribution of this paper in studying the regime when $1\leq n \leq d$. In my understanding, the best batch size in this paper is $O(\log^2(1/\alpha))$ and is made possible when $k=\log(1/\alpha)$ (instead of that $k$ being $\log^2(1/\alpha)$, Line 119?) and this is not yet a constant. Hence, the lower bound $n=1$ of the regime is not reached. I think it is very important to bring this up-to-front when claiming the contribution of the paper to avoid any misunderstanding.
> >
> > Algorithmic approach:
> >
> > The clarification helps a lot. It will be helpful to include it in the revision.

---

> > > ### Author Response · Authors · 2025-04-09
> > >
> > > Thanks for the detailed response. We will clarify the two concerns raised by the reviewer below.
> > >
> > >
> > > **Batch Size Improvement within Polynomial Sample Complexity and Runtime**
> > >
> > > Regarding the phrasing used in the question in Lines 105–107 (“Is there a computationally efficient algorithm for list-decodable linear regression in the batch setting with significantly improved batch size and/or error guarantees?”), the answer is yes—there is indeed an algorithm that offers both quantitative and qualitative improvements in batch size. Specifically, let $\mathcal{A}$ be the algorithm from Theorem 1.3 with $\delta = 1/12$ hardcoded. Then $\mathcal{A}$ runs in polynomial time and requires a batch size of $O(\sqrt{1/\alpha})$, which is a quadratic improvement over the $O(1/\alpha)$ batch size required by previous work. By instantiating the algorithm with a different constant value of $\delta$, we can achieve any desired polynomial improvement in batch size while still maintaining polynomial running time.
> > >
> > > For sub-constant values of $\delta$, the algorithm runs in quasi-polynomial time and achieves even smaller batch size guarantees. In our view, this is orthogonal to the main question in Lines 105–107, which asks whether any algorithm exists that is both efficient and achieves better batch size guarantees than prior work. As such, it does not contradict the answer provided above. Nonetheless, since this point caused confusion for the reviewer, we greatly appreciate their feedback and will clarify it in the final version.
> > >
> > >
> > > **Best-possible Batch Size**
> > >
> > > The reviewer is right that the smallest batch size for our algorithm to work is $\Theta( \log^2(1/\alpha) )$, which corresponds to the parameter $k = \log(1/\alpha)$ (thanks for pointing out the typo!).
> > >
> > > Even though this might suggest that the algorithm cannot work with constant batch size,
> > > and that the upper bound $n = \Theta ( \log^2(1/\alpha) )$ does not immediately match the existing SQ lower bounds from the literature, the claimed hardness is still argued to hold using a reduction from batch to non-batch setting that we mention in line 131 and describe in Appendix F. In particular, the lower bound together with our reduction give strong evidence that the runtime achieved of any algorithm is  **necessary** to be exponential in $1/\alpha$ if the batch size is only $o(\log(1/\alpha))$.  This includes the case when the batch size is any constant, and the number of batches drawn is polynomial. In summary, an important theoretical contribution of our paper is that we essentially characterize the transition point regarding the computational-statistical efficiency of this problem in terms of the batch size parameter, up to a single logarithmic factor.
> > >
> > > We will clarify that our algorithm does not apply for the entire regime of $1 \leq n \leq d$ in the revision.

---

### Decision · Program_Chairs · 2025-05-01

**Decision:**

Accept (poster)

**Comment:**

Overall, the reviewers agree that the key theoretical contributions of the paper are insightful. They are clearly and rigorously stated, and might be of broader interest. Nevertheless, some of the reviewers raised concerns about the problem setting and significance, which should be carefully addressed.